# 2001-2022 global gross primary productivity dataset using an ensemble model based on random forest

Xin Chen[1], Tiexi Chen[1,2,3*], Xiaodong Li[4], Yuanfang Chai[5], Shengjie Zhou[1], Renjie Guo[6], Jie Dai[1]

[1]School of Geographical Sciences, Nanjing University of Information Science and Technology, Nanjing 210044, Jiangsu, China.
[2]Qinghai Provincial Key Laboratory of Plateau Climate Change and Corresponding Ecological and Environmental Effects, Qinghai University of Science and Technology, Xining 810016, China
[3]School of Geographical Sciences, Qinghai Normal University, Xining 810008, Qinghai, China.
[4]Qinghai Institute of Meteorological Science, Xining 810008, Qinghai, China.
[5]Department of Earth Sciences, Vrije Universiteit Amsterdam, Boelelaan 1085, 1081 HV, Amsterdam, the Netherlands
[6]Faculty of Geographical Science, Beijing Normal University, Beijing, China.

*Correspondence to*: Tiexi Chen (txchen@nuist.edu.cn)

**Abstract.** Advancements in remote sensing technology have significantly contributed to the improvement of models for estimating terrestrial gross primary productivity (GPP). However, discrepancies in the spatial distribution and interannual variability within GPP datasets pose challenges to a comprehensive understanding of the terrestrial carbon cycle. In contrast to previous models that rely on remote sensing and environmental variables, we developed an ensemble model based on the random forest (ERF model). This model used GPP outputs from established models (EC-LUE, GPP-kNDVI, GPP-NIRv, Revised-EC-LUE, VPM, MODIS) as inputs to estimate GPP. The ERF model demonstrated superior performance, explaining 85.1% of the monthly GPP variations at 170 sites, surpassing the performance of selected GPP models (67.7%-77.5%) and an independent random forest model using remote sensing and environmental variables (81.5%). Additionally, the ERF model improved accuracy across each month and various subvalues, mitigating the issue of "high value underestimation and low value overestimation" in GPP estimates. Over the period from 2001 to 2022, the global GPP estimated by the ERF model was 132.7 PgC yr$^{-1}$, with an increasing trend of 0.42 PgC yr$^{-2}$, which is comparable to or slightly better than the accuracy of other mainstream GPP datasets in term of validation results of GPP observations independent of FLUXNET (ChinaFlux). Importantly, for the growing number of GPP datasets, our study provides a way to integrate these GPP datasets, which may lead to a more reliable estimate of global GPP.

## 1 Introduction

Gross primary productivity (GPP) is the largest carbon flux in the global carbon cycle, and serves as the primary input of carbon into the terrestrial carbon cycle. Uncertainties in GPP estimates can propagate to other carbon flux estimates, making it crucial to clarify the spatio-temporal patterns of GPP (Xiao et al., 2019; Ruehr et al., 2023). However, global GPP is variously estimated from 90 PgC yr$^{-1}$ to 160 PgC yr$^{-1}$ across different studies, with these variations becoming more pronounced when scaled down to regional scales or specific ecosystem types. This variability underscores the necessity for innovative methods to reduce uncertainty in GPP estimates (Jung et al., 2019; Ryu et al., 2019; Anav et al., 2015).

The light use efficiency (LUE) model is one of the most widely adopted methods for estimating GPP. It assumes that GPP is proportional to the photosynthetically active radiation absorbed by vegetation, and optimizes the spatio-temporal pattern of GPP through meteorological constraints such as temperature and moisture (Pei et al., 2022). However, variations in these constraints varies significantly, leading to differences of over 10% in model explanatory power. (Yuan et al., 2014). Recent studies have proposed some novel vegetation indices that have been shown to be effective proxies for GPP through theoretical derivation and observed validation (Badgley et al., 2017; Camps-Valls et al., 2021). However, these vegetation indices often use only remote sensing data as an input for estimating long-term GPP without considering meteorological factors, which has led to some controversy (Chen et al., 2024; Dechant et al., 2020; Dechant et al., 2022). Both LUE and vegetation index models use linear mathematical formulas to estimate GPP, but ecosystems are inherently complex, and the biases introduced by these numerical models increase the uncertainty of GPP estimates. Machine learning models have shown great potential for improving GPP estimates in previous studies (Jung et al., 2020; Guo et al., 2023). These model are trained by non-physical means directly using GPP observations and selected environmental and vegetation variables, and the performance of the models depends on the number and quality of observed data and the representativeness of input data. Nevertheless, direct validation from flux towers of FLUXNET reveals that these models typically explain only about 70% of monthly GPP variations, with similar performance to other GPP models (Wang et al., 2021; Badgley et al., 2019; Zheng et al., 2020; Jung et al., 2020). Due to deviations in the model structure, a common limitation across these models is the poor estimate of monthly extreme GPP, leading to the phenomenon of "high value overestimation and low value overestimation" (Zheng et al., 2020). Especially for extremely high values, which usually occur during the growing season and largely determine the annual totals and interannual fluctuations of GPP, this underestimation may hinder our understanding of the global carbon cycle.

It is challenging for a single model to provide accurate estimates for all global regions. Ensemble models have outperformed individual models in previous studies, potentially addressing some inherent issues in model estimate (Chen et al., 2020; Yao et al., 2014). Traditional multi-model ensemble methods usually use a simple multi-model average or a weighted Bayesian average. However, these methods typically assign fixed weights to each model and are essentially linear combinations. Recent studies have incorporated machine learning techniques to multi-model ensembles to establish nonlinear relationships between multiple simulated target variables and real target variable, improving simulation performance (Bai et al., 2021; Yao et al.,

2017; Tian et al., 2023). Whether this method can improve some common problems with individual GPP models, such as high
value underestimation and low value overestimation, is not clear and needs to further investigation.
In this study, we attempt to use an ensemble model based on the random forest (ERF model) to improve global GPP estimate.
Specifically, the work of this study includes the following: (1) Recalibrating parameters for each model, and comparing the
performance of six GPP models and the ERF model; (2) Focusing on the phenomenon of "high value underestimation and low
value overestimation" in each model, and evaluating the performance of each model across different months, vegetation types
and subvalues (high value, median value, low value); (3) Developing a global GPP dataset using the ERF model and validating
its generalization using GPP observations from ChinaFlux.

## 71 2 Method

### 72 2.1 Data at the global scale

In this study, we selected remote sensing data from the Moderate Resolution Imaging Spectroradiometer (MODIS) and
meteorological data from EAR5 to estimate global GPP (Hersbach et al., 2020). For the remote sensing data, surface reflectance
(red band, near infrared band, blue band and shortwave infrared band), leaf area index (LAI) and fraction of photosynthetically
active radiation (FPAR) were used. For meteorological data, we selected average air temperature, dew point temperature,
minimum air temperature, total solar radiation and direct solar radiation. Dew point temperature and air temperature were used
to calculate saturated vapor pressure difference (VPD) (Yuan et al., 2019), and diffuse solar radiation (DifSR) was derived as
the difference between total solar radiation and direct solar radiation. Minimum air temperature was obtained from the hourly
air temperature. $CO_2$ data were obtained from the monthly average carbon dioxide levels measured by the Mauna Loa
Observatory in Hawaii. Table 1 provides an overview of the datasets used in this study.

**Table 1.** Overview of the datasets used in this study.

| Variable | Dataset | Spatial resolution | Temporal resolution | Temporal coverage |
|---|---|---|---|---|
| Surface reflectance (red band and near infrared band) | MCD43C4 | 0.05 ° | daily | 2001-2022 |
| Surface reflectance (red band, near infrared band, blue band and shortwave infrared band) | MOD09CMG | 0.05 ° | daily | 2001-2022 |
| LAI | MOD15A2H | 500m | 8d | 2001-2022 |
| FPAR | MOD15A2H | 500m | 8d | 2001-2022 |
| Average air temperature (AT) | ERA5-land | 0.1 ° | Monthly | 2001-2022 |
| Dew point temperature (DPT) | ERA5-land | 0.1 ° | Monthly | 2001-2022 |

| | | | | |
|---|---|---|---|---|
| Minimum air temperature (MINT) | ERA5-land | 0.1 ° | Monthly | 2001-2022 |
| Total solar radiation (TSR) | ERA5 monthly data on single levels | 0.25 ° | Monthly | 2001-2022 |
| Direct solar radiation (DirSR) | ERA5 monthly data on single levels | 0.25 ° | Monthly | 2001-2022 |
| $CO_2$ | NOAA's Earth System Research Laboratory | / | Monthly | 2001-2022 |
| Distribution map of C4 crops | Harvested Area and Yield for 175 Crops | 1/12 ° | Annual | 2000 |
| Land use | MCD12C1 | 0.05 ° | Annual | 2010 |


Previous studies have shown that the photosynthetic capacity of C4 crops is much higher than that of C3 crops (Chen et al.,
2014; Chen et al., 2011), so it is necessary to divide the cropland into C3 crops and C4 crops. To estimate the global GPP, we
used the "175 Crop harvested Area and yield" dataset, which describes the global harvested area and yield of 175 crops in
2000 (Monfreda et al., 2008). We extracted the sum of the area ratios of all C4 crops (corn, corn feed, sorghum, sorghum feed,
sugarcane, millet) at each grid as the coverage of C4 crops (Figure S1). Consequently, the estimated value of cropland GPP
can be expressed as: coverage of C3 crops × simulated GPP value of C3 crops + coverage of C4 crops × simulated GPP value
of C4 crops, which has been used in a previous study (Guo et al., 2023).
The land use map was derived from the IGBP classification of MCD12C1, and 2010 was chosen as the reference year (that is,
land use data is unchanged in the simulation of global GPP). In order to meet the requirements of subsequent research, land
cover types were grouped into 9 categories: Deciduous Broadleaf Forest (DBF), Evergreen Needleleaved Forest (ENF),
Evergreen Broadleaf Forest (EBF), Mixed Forest (MF), Grassland (GRA), Cropland (including CRO-C3 and CRO-C4),
Savannah (SAV), Shrub (SHR), Wetland (WET).
Finally, for higher resolution data, we gridded the dataset to 0.05 ° by averaging all pixels whose center fell within each 0.05 °
grid cell for upscaling. For lower resolution data, we used the nearest neighbor resampling method to 0.05 °. In addition,
MODIS data were aggregated to a monthly scale to ensure spatio-temporal consistency.
**2.2 Observation data at the site scale**
GPP observations were sourced from the FLUXNET 2015 dataset, which includes carbon fluxes and meteorological variables
from more than 200 flux sites around the world (Pastorello et al., 2020). GPP cannot be obtained directly from flux sites and
usually needs to be obtained by dismantling the Net Ecosystem Exchange. We chose a monthly level GPP based on the
nighttime partitioning method and retained only high quality data (NEE_VUT_REF_QC > 0.8) for every year, ultimately
selecting 170 sites with 10932 monthly values for this study. In addition, we selected monthly average air temperature, total
solar radiation and VPD. The site observations do not provide direct solar radiation, so we extracted data from ERA5 covering
the flux tower. Monthly minimum air temperature was derived from hourly air temperature. Since some required model data
are not directly available at flux sites, LAI and FPAR were extracted from MOD15A2H (500 m), and surface reflectance data
(red band, near infrared band, blue band and shortwave infrared band) were derived from MCD43A4 (500 m) and MOD09A1
(500 m). These data are roughly similar to the footprint of the flux site and can represent the land surface of the site (Chu et
al., 2021).

## 112  2.3 GPP estimate model

We selected six independent models to estimate GPP in this study. These models are widely used with few model parameters
and have demonstrated reliable accuracy in previous studies (Zheng et al., 2020; Zhang et al., 2017; Badgley et al., 2017). The
six models are EC-LUE, Revised-EC-LUE, NIRv-based linear model, kNDVI-based linear model, VPM, MODIS. The VPM,
MODIS and EC-LUE are LUE models based on remote sensing data and meteorological data (Yuan et al., 2007; Running et
al., 2004; Xiao et al., 2004). Zheng et al., (2020) proposed the Revised-EC-LUE model, which divides the canopy into sunlit
and shaded leaves, improving the estimate of global GPP (Zheng et al., 2020). The NIRv and kNDVI are novel vegetation
indices calculated from the red and near-infrared bands of the reflectance spectrum (Badgley et al., 2017; Camps-Valls et al.,
2021). Similar to solar induced chlorophyll fluorescence, they exhibit a linear relationship with GPP and are considered
effective proxies for GPP. Detailed descriptions of all models can be found in Text S1.
To reduce uncertainty in GPP estimates from a single model, we used the ERF model, the basic idea of which is to restructure
the simulated values of multiple models. In this study, we directly used the ERF model to establish the relationship between
the GPP simulated by the above six models and GPP observations. In addition, for comparison with the ERF model, we also
used the random forest (RF) method for modeling. In this study, we used average air temperature, minimum air temperature,
VPD, direct solar radiation, diffuse solar radiation, FPAR and LAI to estimate GPP. Both models used the random forest
method, which has been widely used in previous studies of GPP estimate (Jung et al., 2020; Guo et al., 2023). Random forest
is an ensemble learning algorithm that combines the outputs of multiple decision trees to produce a single result, and is
commonly used for classification and regression problems (Belgiu and Drăguţ, 2016). In the regression problem, the output
result of each decision tree is a continuous value, and the average of all decision tree outputs is taken as the final result. An
overview of all models used can be found in Table 2.
**Table 2.** Overview of the models used in this study.

| ID | Model | Input data | Output |
|---|---|---|---|
| 1 | EC-LUE | FPAR, VPD, AT, SRAD, $CO_2$ | $GPP_{EC}$ |

| 2 | Revised-EC-LUE | LAI, VPD, AT, DifSR, DirSR, $CO_2$ | $GPP_{REC}$ |
|---|---|---|---|
| 3 | kNDVI-GPP | Red band and near infrared band (MCD43) | $GPP_{kNDVI}$ |
| 4 | NIRv-GPP | Red band and near infrared band (MCD43) | $GPP_{NIRv}$ |
| 5 | VPM | Red band, near infrared band, blue band, shortwave infrared band (MOD09), AT, SRAD | $GPP_{VPM}$ |
| 6 | MODIS | FPAR, SRAD, MINT, VPD | $GPP_{MODIS}$ |
| 7 | Random forest model (RF) | LAI, FPAR, AT, MINT, VPD, DifSR, DirSR | $GPP_{RF}$ |
| 8 | Ensemble model based on random forest (ERF) | $GPP_{EC}$, $GPP_{REC}$, $GPP_{kNDVI}$, $GPP_{NIRv}$, $GPP_{MODIS}$, $GPP_{VPM}$ | $GPP_{ERF}$ |


## 2.4 Model parameter calibration and validation

FLUXNET only provides GPP observations and meteorological data, lacking direct measurements for LAI, FPAR, and surface
reflectance, so remote sensing data is needed. Considering the variety of remote sensing data sources, such as MODIS and
AVHRR, it is evident that calibrating the same GPP model with different remote sensing data can yield varied parameters. In
addition, the number of sites used to calibrate model parameters is also an important influencing factor for model parameters.
The original parameters of these models were calibrated with only a limited number of sites (e.g., 95 sites for Revised EC-
LUE and 104 for NIRv) (Wang et al., 2021; Zheng et al., 2020). Therefore, to reduce the impact of the uncertainty of model
parameters on simulation results, we did not use original parameters and conducted parameter calibration for GPP models
across different vegetation types. For EC-LUE, Revised EC-LUE, VPM and MODIS, the Markov chain Monte Carlo method
was used to calibrate model parameters. Traditionally, the mean of the posterior distribution of parameters is taken as the
optimal value. However, previous studies have indicated that some model parameters are not well constrained when calibrating
multiple model parameters (Xu et al., 2006; Wang et al., 2017), so we selected the parameter with the smallest root-mean-
square error (RMSE) as the optimal parameter in each iteration. For each vegetation type, we randomly selected 70% of the
sites for parameter calibration, and repeated the process 200 times. In order to avoid overfitting, we adopted the mean of the
200 calibrated parameters as the final model parameters. Similarly, for the two vegetation index models, we randomly selected
70% of the sites in each vegetation type for parameter calibration, repeating the process 200 times and using the mean of the
200 calibrated parameters as the final model parameters.
After obtaining GPP estimates from the six GPP models, we evaluated the simulation performance of the RF model and the
ERF model respectively. For both models, we evaluated the model performance using 5-fold cross-validation, where the
process was repeated 200 times, and the mean of the 200 GPP estimates was considered the final GPP estimate. In addition,
we used a second validation method where 70% of the data was selected for modeling and only the remaining 30% was
validated, a process that was repeated 200 times. We utilized the determination coefficient ($R^2$) and RMSE as metrics to
evaluate the simulation performance of all models. Additionally, we used the ratio of GPP simulations to GPP observations
(Sim/Obs) to measure whether the model overestimates or underestimates.

## 2.5 Global GPP estimate based on ERF model and its uncertainty.

Based on the ERF model, we estimated global GPP for 2001-2022 (ERF_GPP). It is important to note that in this process, we
used all the site data to build the model. The uncertainties of ERF_GPP can be attributed to two primary factors: the influence
of the number of GPP observations and the influence of the number of features (that is, the simulated GPP). For the first type
of uncertainty, we randomly selected 80% of the data to build a model and simulate the multi-year average of global GPP. The
process was repeated 100 times, yielding 100 sets of multi-year averages of ERF_GPP. Their standard deviations were
considered as the uncertainty of ERF_GPP caused by the number of GPP observations. For the second type of uncertainty, we
selected different number of features to build a model and simulate the multi-year average of global GPP. A total of 56 sets of
multi-year averages of ERF_GPP were obtained. The standard deviation of different combinations was considered to be the
uncertainty of ERF_GPP caused by the number of features.

## 2.6 Evaluation of the generalization of different GPP datasets

The majority of flux sites in FLUXNET are concentrated in Europe and North America, it is unclear whether the different GPP
estimate methods are suitable for regions with sparse flux sites. Recently, ChinaFlux has published GPP observations from
several sites, offering an opportunity to evaluate the generalization of different GPP datasets. However, the spatial resolution
of most GPP datasets is 0.05 °, and a direct comparison with GPP observations at flux sites is challenging. Therefore, we
extracted 0.05 °MODIS land use covering the flux sites. If the vegetation type of the flux site matched the MODIS land use,
the site was used for the analysis. Finally, a total of 12 flux sites were selected (Figure S2), and Table S1 shows the information
of these sites. The same procedure was applied to FLUXNET, resulting in the selection of 52 sites (Figure S2). It should be
noted that due to the absence of meteorological data from some sites in Chinaflux, we did not validate all GPP models at the
site scale (500 m).
We evaluated the generalization of ERF_GPP at 12 ChinaFlux sites and 52 FLUXNET sites. In addition, we selected a number
of widely used GPP datasets for comparison, including BESS (Li et al., 2023), GOSIF (Li and Xiao, 2019), FLUXCOM:
random forest-based version (FLUXCOM-RF) and ensemble version (FLUXCOM-ENS) (Jung et al., 2020), NIRv (Wang et
al., 2021), Revise-EC-LUE (Zheng et al., 2020), MODIS (Running et al., 2004), VPM (Zhang et al., 2017), which were
generated using different GPP estimate methods. These GPP datasets all have a spatial resolution of 500 m-0.5 °, similar to the
resampling process in section 2.1, we have unified them to 0.05 °. The common time range for these datasets spanned from
2001 to 2018, and the temporal resolution was unified to monthly to match the GPP observations.

## 3 Result

### 3.1 Performance of GPP models at site scale

Table S2-S7 show the optimization results of the six GPP model parameters. Consistent with previous study, in the Revised EC-LUE model, the light use efficiency parameter of shade leaves was significantly higher than that of sunlit leaves (Zheng et al., 2020). It is necessary to divide cropland into C3 crops and C4 crops. In all models, the light use efficiency parameters of C4 crops were significantly higher than those of C3 crops, which was particularly reflected in the two vegetation index models of $GPP_{kNDVI}$ and $GPP_{NIRv}$, the slope of the linear regression directly reflected the difference in photosynthetic capacity of the different crops.

Figure 1 shows the performance of all models across different vegetation types. Overall, the performance of the ERF model was better than that of the other GPP models. $GPP_{ERF}$ had the higher accuracy among all models, with $R^2$ between 0.61-0.91 and RMSE between 0.72-2.78 gC m$^{-2}$ d$^{-1}$. In contrast, the LUE and vegetation index models performed slightly weaker, especially in EBF, where $R^2$ was both below 0.5. It is worth noting that compared to other vegetation types, the RMSE was highest for cropland, with 6 out of 8 models for C4 crop exceeding 3 gC m$^{-2}$ d$^{-1}$, suggesting that these existing GPP models may not properly capture the seasonal changes in cropland GPP. The six models with calibration parameters and ERF model were found to have no significant deviation across vegetation types. However, $GPP_{RF}$ was significantly underestimated for C4 crops and overestimated for SHR.

**a**

| | DBF | ENF | EBF | MF | GRA | CRO-C3 | CRO-C4 | SAV | SHR | WET |
|---|---|---|---|---|---|---|---|---|---|---|
| $GPP_{EC}$ | 0.82 | 0.8 | 0.36 | 0.8 | 0.78 | 0.62 | 0.77 | 0.72 | 0.74 | 0.7 |
| $GPP_{NIRv}$ | 0.87 | 0.7 | 0.25 | 0.77 | 0.79 | 0.64 | 0.8 | 0.86 | 0.69 | 0.6 |
| $GPP_{kNDVI}$ | 0.85 | 0.6 | 0.23 | 0.71 | 0.75 | 0.67 | 0.79 | 0.79 | 0.64 | 0.56 |
| $GPP_{REC}$ | 0.84 | 0.81 | 0.44 | 0.79 | 0.82 | 0.66 | 0.78 | 0.78 | 0.8 | 0.68 |
| $GPP_{VPM}$ | 0.89 | 0.77 | 0.22 | 0.79 | 0.82 | 0.72 | 0.89 | 0.86 | 0.79 | 0.75 |
| $GPP_{MODIS}$ | 0.71 | 0.8 | 0.27 | 0.74 | 0.69 | 0.56 | 0.52 | 0.79 | 0.7 | 0.73 |
| $GPP_{RF}$ | 0.89 | 0.86 | 0.6 | 0.84 | 0.84 | 0.68 | 0.85 | 0.87 | 0.8 | 0.74 |
| $GPP_{ERF}$ | 0.91 | 0.86 | 0.61 | 0.83 | 0.87 | 0.74 | 0.87 | 0.89 | 0.85 | 0.74 |

**b**

$gC\ m^{-2}\ d^{-1}$

| | DBF | ENF | EBF | MF | GRA | CRO-C3 | CRO-C4 | SAV | SHR | WET |
|---|---|---|---|---|---|---|---|---|---|---|
| $GPP_{EC}$ | 2 | 1.54 | 2.69 | 1.57 | 1.87 | 2.63 | 4.2 | 1.38 | 0.97 | 1.9 |
| $GPP_{NIRv}$ | 1.7 | 1.85 | 2.72 | 1.68 | 1.82 | 2.53 | 3.54 | 0.9 | 1.04 | 2.23 |
| $GPP_{kNDVI}$ | 1.8 | 2.08 | 2.76 | 1.87 | 1.94 | 2.39 | 3.3 | 1.08 | 1.1 | 2.31 |
| $GPP_{REC}$ | 1.9 | 1.53 | 2.45 | 1.66 | 1.67 | 2.45 | 3.89 | 1.16 | 0.85 | 1.97 |
| $GPP_{VPM}$ | 1.56 | 1.95 | 3.29 | 1.93 | 1.66 | 2.18 | 2.5 | 0.91 | 0.84 | 1.78 |
| $GPP_{MODIS}$ | 2.58 | 1.51 | 2.91 | 1.88 | 2.17 | 2.77 | 5.1 | 1.12 | 1.02 | 1.79 |
| $GPP_{RF}$ | 1.61 | 1.24 | 1.98 | 1.53 | 1.57 | 2.37 | 3.81 | 0.85 | 1.19 | 1.91 |
| $GPP_{ERF}$ | 1.4 | 1.24 | 1.97 | 1.46 | 1.38 | 2.15 | 2.78 | 0.81 | 0.72 | 1.78 |

**c**

| | DBF | ENF | EBF | MF | GRA | CRO-C3 | CRO-C4 | SAV | SHR | WET |
|---|---|---|---|---|---|---|---|---|---|---|
| $GPP_{EC}$ | 1.06 | 0.96 | 0.96 | 0.96 | 1 | 1 | 1 | 1.03 | 1.18 | 1.01 |
| $GPP_{NIRv}$ | 1.03 | 1.04 | 1.01 | 1 | 1.04 | 1.07 | 1.11 | 1 | 1.06 | 1.08 |
| $GPP_{kNDVI}$ | 1 | 1 | 1.01 | 1 | 1 | 1.02 | 1.03 | 1.01 | 1 | 1.02 |
| $GPP_{REC}$ | 1.05 | 0.97 | 0.98 | 0.96 | 1.02 | 1.04 | 1.08 | 1.02 | 1.12 | 1.02 |
| $GPP_{VPM}$ | 0.96 | 0.99 | 0.95 | 0.99 | 0.97 | 1.03 | 1.01 | 1 | 0.98 | 0.98 |
| $GPP_{MODIS}$ | 1.03 | 0.95 | 0.96 | 0.99 | 1 | 1.08 | 0.95 | 1.04 | 1.04 | 0.96 |
| $GPP_{RF}$ | 1.04 | 0.96 | 1.01 | 1.08 | 0.98 | 1 | 0.72 | 0.97 | 1.26 | 1.18 |
| $GPP_{ERF}$ | 1.03 | 0.98 | 1.01 | 0.98 | 0.99 | 1.01 | 1.07 | 0.98 | 0.95 | 1 |


**Figure 1.** The performance of the eight models on different vegetation types. a, b and c represent $R^2$, RMSE, and Sim/Obs respectively.
Combining the results of all flux sites, $GPP_{ERF}$ explained 85.1% of the monthly GPP variations, while the seven GPP models
only explained 67.7%-81.5% of the monthly GPP variations (Figure 2). Another validation method also showed similar results
(Figure S3). In order to further prove the robustness of the ERF model, we also used GPP models with original parameters for
modeling and validation. As shown in Figure S4, the performance of these GPP models decreased significantly, with $R^2$ ranging
from 0.570 to 0.719 and RMSE ranging from 2.29 to 3.81 gC m$^{-2}$ d$^{-1}$. The phenomenon of "high value underestimation and
low value overestimation" was also pronounced. However, the ERF model maintained a consistent advantage, with $R^2$
significantly higher than other GPP models (0.856). In addition, we tested the effect of the number of GPP models on the
accuracy of the ERF model. As shown in Table S8, as the number of GPP in the ERF model increased, the performance gain
of the model gradually decreased.
In summary, $GPP_{ERF}$ showed high accuracy in terms of vegetation type and the ability to interpret monthly variations in GPP,
which also illustrates the potential of the ERF model to improve GPP estimate. However, it was observed that most GPP
simulations exhibited the phenomenon of "high value underestimation and low value overestimation". For example, $GPP_{EC}$,
$GPP_{REC}$, $GPP_{MODIS}$ and $GPP_{RF}$ showed obvious underestimation in the months when the monthly GPP value surpassed 15 gC
m$^{-2}$ d$^{-1}$ (Figure 2). Therefore, it is necessary to evaluate the performance of different models in each month and different
subvalues.

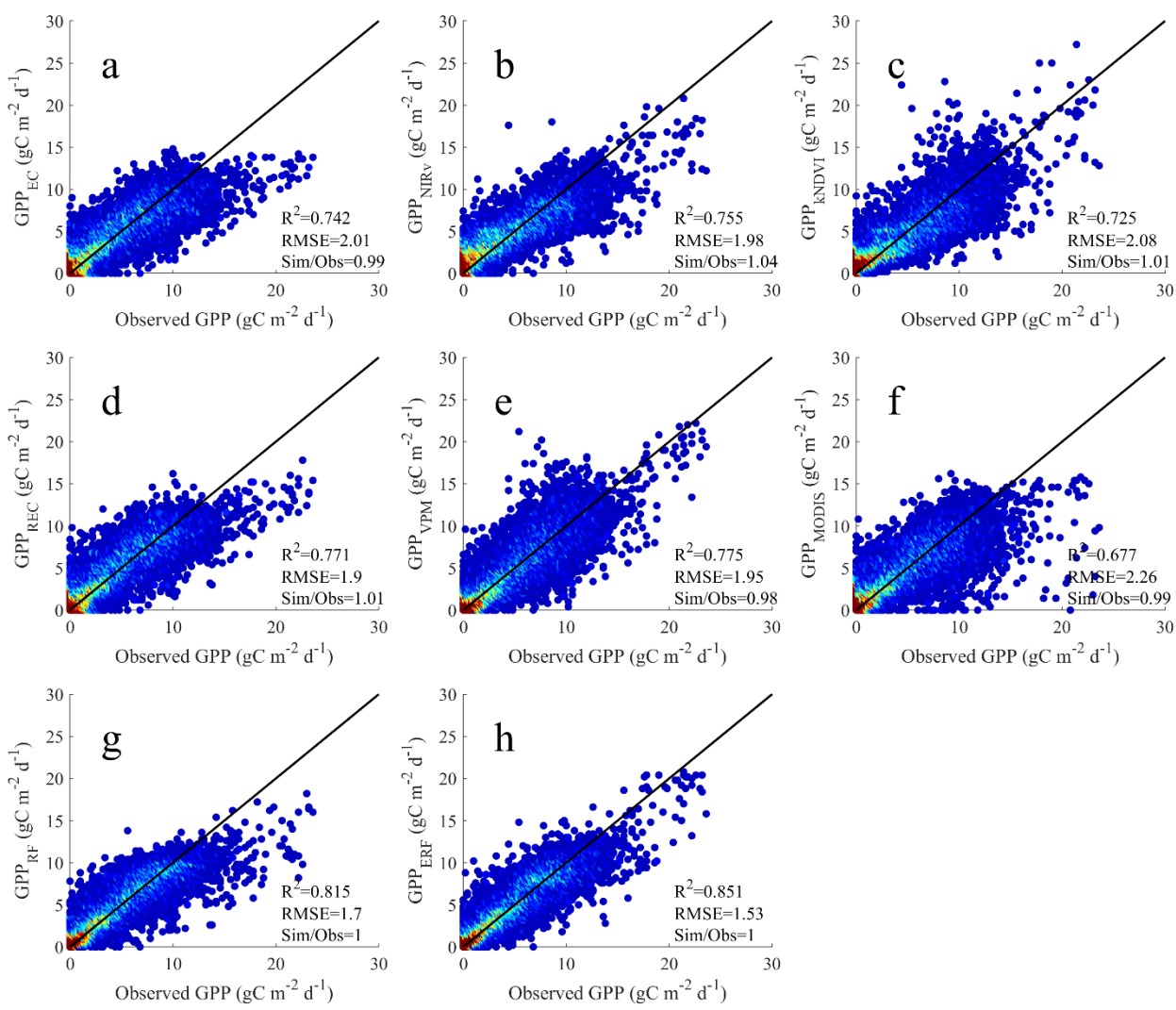


**Figure 2.** Comparison between the GPP simulations of the eight models and the GPP observations. a-h represents GPP$_{EC}$, GPP$_{NIRv}$, GPP$_{kNDVI}$, GPP$_{REC}$, GPP$_{VPM}$, GPP$_{MODIS}$, GPP$_{RF}$, GPP$_{ERF}$, respectively.


## 3.2 Performance of GPP models in each month and different subvalues

Figure 3 shows the simulation accuracy of the eight models in each month. The ERF model maintained a higher accuracy than other GPP models, with GPP$_{ERF}$ consistently achieving higher $R^2$ and lower RMSE in most months, and no evident phenomenons of "high value underestimation and low value overestimation". In contrast, the accuracy of other GPP models was less satisfactory accuracy, especially during winter (most flux sites are concentrated in the Northern Hemisphere), the

LUE models tended to underestimate GPP, and the Sim/Obs remained at 0.72-1.01, although $R^2$ were above 0.7. Meanwhile,
the vegetation index models overestimated GPP, Sim/Obs remained at 1.34-1.73, and $R^2$ were relatively low, mostly around

229 0.6.

## a

| | Jan. | Feb. | Mar. | Apr. | May | Jun. | Jul. | Aug. | Sep. | Oct. | Nov. | Dec. |
|---|---|---|---|---|---|---|---|---|---|---|---|---|
| GPP$_{EC}$ | 0.78 | 0.73 | 0.67 | 0.53 | 0.49 | 0.63 | 0.62 | 0.61 | 0.62 | 0.63 | 0.73 | 0.81 |
| GPP$_{NIRv}$ | 0.61 | 0.7 | 0.73 | 0.64 | 0.65 | 0.72 | 0.73 | 0.7 | 0.64 | 0.6 | 0.56 | 0.53 |
| GPP$_{kNDVI}$ | 0.63 | 0.64 | 0.65 | 0.6 | 0.63 | 0.66 | 0.65 | 0.61 | 0.58 | 0.62 | 0.63 | 0.56 |
| GPP$_{REC}$ | 0.81 | 0.78 | 0.72 | 0.58 | 0.56 | 0.65 | 0.66 | 0.65 | 0.64 | 0.67 | 0.78 | 0.84 |
| GPP$_{VPM}$ | 0.81 | 0.77 | 0.72 | 0.58 | 0.64 | 0.66 | 0.64 | 0.6 | 0.56 | 0.65 | 0.79 | 0.82 |
| GPP$_{MODIS}$ | 0.74 | 0.72 | 0.66 | 0.47 | 0.42 | 0.52 | 0.42 | 0.43 | 0.46 | 0.57 | 0.7 | 0.78 |
| GPP$_{RF}$ | 0.88 | 0.85 | 0.78 | 0.64 | 0.65 | 0.71 | 0.67 | 0.67 | 0.69 | 0.77 | 0.85 | 0.88 |
| GPP$_{ERF}$ | 0.87 | 0.88 | 0.83 | 0.69 | 0.71 | 0.77 | 0.79 | 0.74 | 0.7 | 0.77 | 0.87 | 0.9 |

## b

gC m$^{-2}$ d$^{-1}$

| | Jan. | Feb. | Mar. | Apr. | May | Jun. | Jul. | Aug. | Sep. | Oct. | Nov. | Dec. |
|---|---|---|---|---|---|---|---|---|---|---|---|---|
| GPP$_{EC}$ | 1.25 | 1.36 | 1.51 | 2.21 | 2.68 | 2.56 | 3.02 | 2.45 | 1.81 | 1.45 | 1.14 | 1.09 |
| GPP$_{NIRv}$ | 1.77 | 1.54 | 1.37 | 1.88 | 2.25 | 2.36 | 2.61 | 2.15 | 1.74 | 1.81 | 1.85 | 1.98 |
| GPP$_{kNDVI}$ | 1.75 | 1.71 | 1.56 | 2.02 | 2.35 | 2.57 | 2.86 | 2.57 | 1.84 | 1.51 | 1.55 | 1.87 |
| GPP$_{REC}$ | 1.15 | 1.26 | 1.39 | 2.09 | 2.56 | 2.46 | 2.8 | 2.31 | 1.78 | 1.37 | 1.05 | 1 |
| GPP$_{VPM}$ | 1.2 | 1.29 | 1.45 | 2.05 | 2.27 | 2.58 | 2.93 | 2.59 | 1.89 | 1.42 | 1.06 | 1.11 |
| GPP$_{MODIS}$ | 1.31 | 1.38 | 1.54 | 2.27 | 2.88 | 2.92 | 3.59 | 2.99 | 2.12 | 1.51 | 1.2 | 1.16 |
| GPP$_{RF}$ | 0.89 | 1.02 | 1.22 | 1.84 | 2.21 | 2.23 | 2.7 | 2.24 | 1.54 | 1.1 | 0.86 | 0.85 |
| GPP$_{ERF}$ | 0.92 | 0.92 | 1.08 | 1.71 | 2.01 | 1.97 | 2.16 | 1.99 | 1.59 | 1.12 | 0.8 | 0.8 |

## c

| | Jan. | Feb. | Mar. | Apr. | May | Jun. | Jul. | Aug. | Sep. | Oct. | Nov. | Dec. |
|---|---|---|---|---|---|---|---|---|---|---|---|---|
| GPP$_{EC}$ | 0.78 | 0.86 | 1.04 | 1.17 | 1.08 | 0.94 | 0.88 | 0.97 | 1.13 | 1.12 | 0.96 | 0.84 |
| GPP$_{NIRv}$ | 1.49 | 1.34 | 1.12 | 0.93 | 0.91 | 0.87 | 0.88 | 0.95 | 1.11 | 1.39 | 1.72 | 1.73 |
| GPP$_{kNDVI}$ | 1.55 | 1.4 | 1.11 | 0.86 | 0.89 | 0.9 | 0.9 | 0.92 | 0.99 | 1.18 | 1.5 | 1.69 |
| GPP$_{REC}$ | 0.8 | 0.84 | 1 | 1.17 | 1.12 | 0.97 | 0.91 | 0.98 | 1.13 | 1.1 | 0.96 | 0.86 |
| GPP$_{VPM}$ | 0.72 | 0.77 | 0.81 | 0.88 | 1 | 1.06 | 1.08 | 1.06 | 1 | 0.86 | 0.77 | 0.74 |
| GPP$_{MODIS}$ | 0.87 | 0.96 | 1.09 | 1.09 | 1.03 | 0.95 | 0.91 | 0.98 | 1.07 | 1.05 | 1.01 | 0.92 |
| GPP$_{RF}$ | 0.98 | 1.02 | 1.03 | 1.04 | 1.02 | 0.98 | 0.95 | 0.99 | 1.01 | 1.03 | 1.07 | 1.04 |
| GPP$_{ERF}$ | 0.98 | 0.97 | 0.96 | 0.96 | 1.01 | 0.97 | 0.96 | 1.01 | 1.08 | 1.08 | 1.07 | 1.03 |


**Figure 3.** Performance of the eight models in each month. a, b and c represent $R^2$, RMSE, and Sim/Obs respectively.
We compared the performance of all models in different subvalues, including high value (GPP > 15 gC m$^{-2}$ d$^{-1}$), median value
(15 gC m$^{-2}$ d$^{-1}$ > GPP > 2 gC m$^{-2}$ d$^{-1}$), low value (GPP < 2 gC m$^{-2}$ d$^{-1}$). For extreme values, most models performed poorly
(Figure 4), with $R^2$ for GPP models falling below 0.3, and only GPP$_{VPM}$ showing better performance in the high value. GPP$_{ERF}$
demonstrated some improvement in both low and high values, with $R^2$ 0.32 and 0.43, RMSE of 0.89 and 4.73 gC m$^{-2}$ d$^{-1}$, and
Sim/Obs closer to 1, respectively. In the median value, all models performed better, with no significant bias in the GPP estimate.
The $R^2$ of GPP models ranged from 0.44 to 0.68, and the RMSE remained between 1.82 and 2.54 gC m$^{-2}$ d$^{-1}$. Further analysis
was made at two typical sites, it was obvious that GPP$_{EC}$, GPP$_{REC}$ and GPP$_{MODIS}$ on CN-Qia exhibited obvious underestimation
during the growing season (Figure S5). On CH_Lae, GPP$_{kNDVI}$ and GPP$_{VPM}$ were significantly overestimated (Figure S6). In
contrast, at both sites, GPP$_{ERF}$ was more consistent with observations, indicating that the superior performance of GPP$_{ERF}$ was
due to the corrections on the time series.

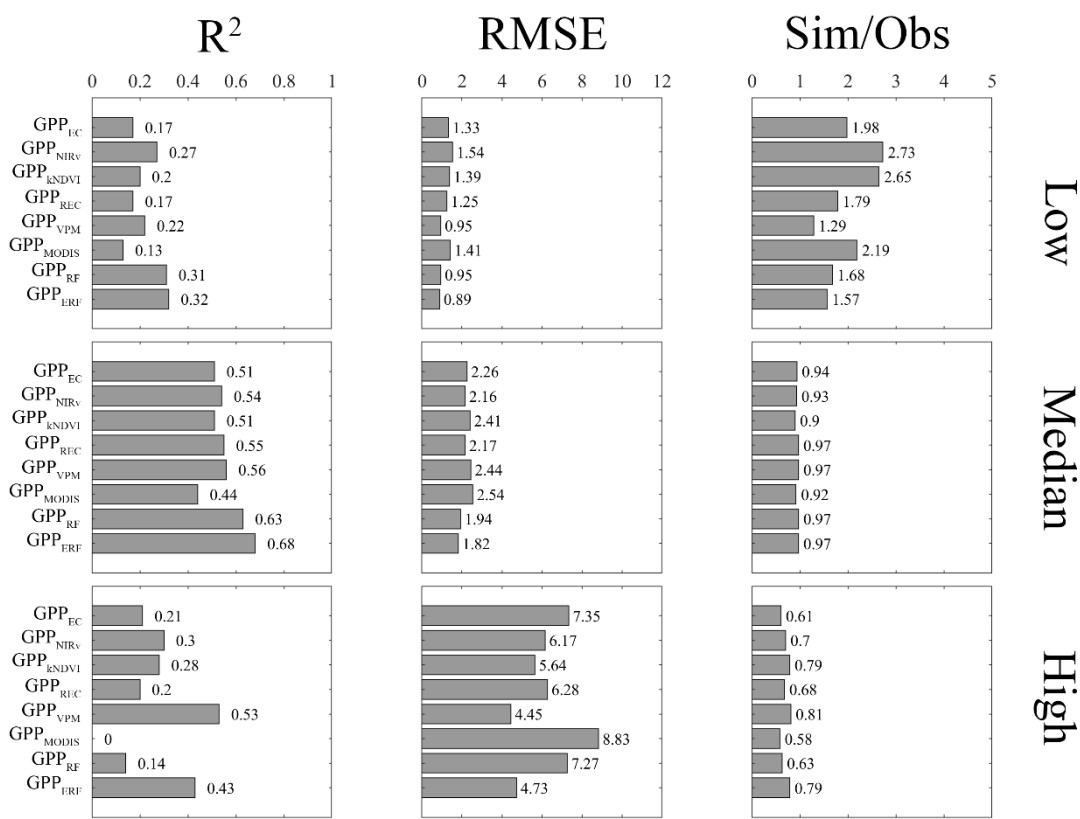

**Figure 4.** Performance of eight models in different subvalues.

**3.3 Temporal and spatial characteristics of ERF GPP and its generalization evaluation**

Figure 5a shows the spatial distribution of the multi-year average of ERF_GPP. The high values of GPP were mainly concentrated in tropical areas, exceeding 10 gC m$^{-2}$ d$^{-1}$, and relatively high in southeastern North America, Europe and southern China, about 4-6 gC m$^{-2}$ d$^{-1}$. From 2001-2022, China and India showed the fastest increase in GPP, mostly at 0.1 gC m$^{-2}$ d$^{-1}$ (Figure 5b), similar to a previous study that reported that China and India led the global greening (Chen et al., 2019). We further investigated the annual maximum GPP, as shown in Figure 5c, and the North American corn belt was the global leader in GPP at more than 15 gC m$^{-2}$ d$^{-1}$, compared to only 10 gC m$^{-2}$ d$^{-1}$ in most tropical forests. In 2001-2022, the global GPP was 132.7 $\pm$2.8 PgC yr$^{-1}$, with an increasing trend of 0.42 PgC yr$^{-2}$ (Figure 5d). The lowest value was 128.6 PgC yr$^{-1}$ in 2001, and the highest value was 136.2 PgC yr$^{-1}$ in 2020.

The results of the two uncertainty analyses consistently indicated that ERF_GPP exhibited higher uncertainty in tropical regions (Figures S7 and S8), and the uncertainty of ERF_GPP caused by the number of GPP observations was relatively small, the standard deviation of 100 simulations was about 0.3 gC m$^{-2}$ d$^{-1}$ in the tropics and lower in other regions, below 0.1 gC m$^{-2}$ d$^{-1}$. In contrast, the uncertainty of ERF_GPP caused by the number of features was more pronounced, especially when fewer features were included in the models. It is worth noting that when the number of features was five, the uncertainty was already substantially less, and the standard deviation was generally lower than 0.5 gC m$^{-2}$ d$^{-1}$.

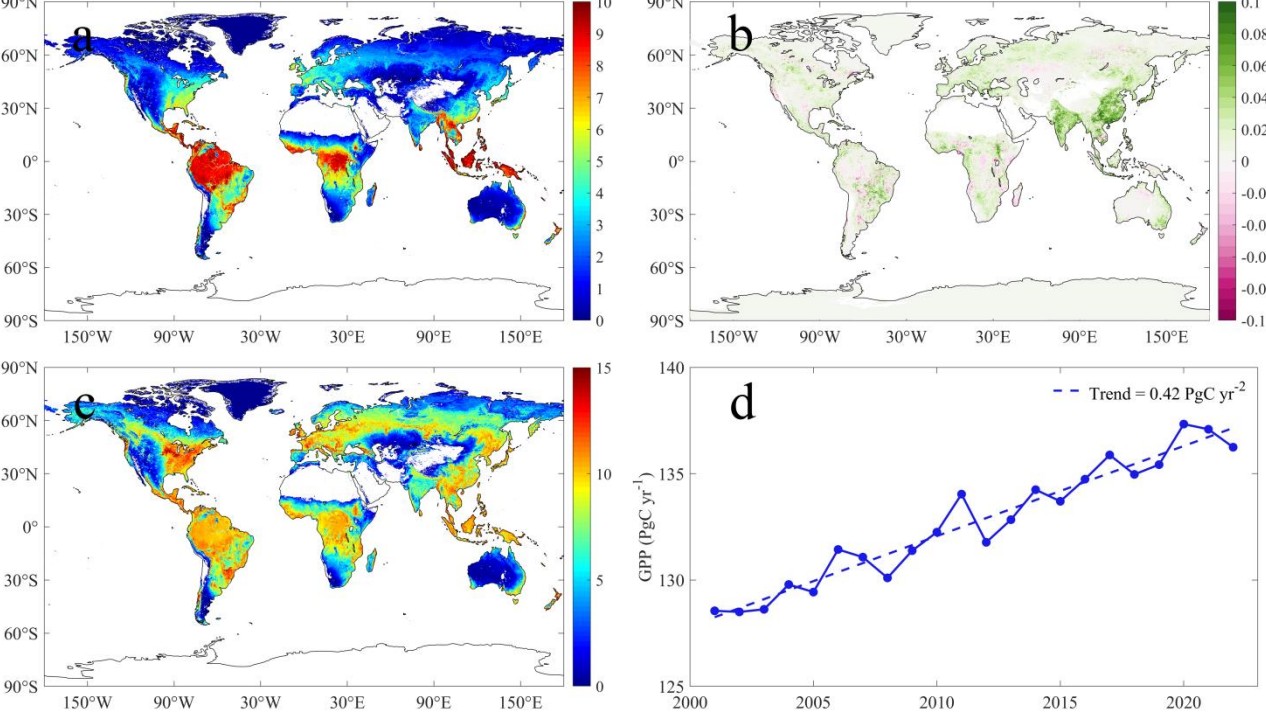

**Figure 5.** Spatial distribution and interannual changes of ERF_GPP during 2001-2022. a represents the multi-year average, b represents the
trend, c represents the annual maximum, and d represents the interannual change of GPP.

As shown in Figure 6, ERF_GPP and other GPP datasets were validated using GPP observations from ChinaFlux. Among all
models, $GPP_{VPM}$ demonstrated the best performance, with $R^2$ of 0.86 and RMSE of 1.34 gC m$^{-2}$ d$^{-1}$. ERF_GPP also exhibited
high generalization, with $R^2$ of 0.75, RMSE of 1.72 gC m$^{-2}$ d$^{-1}$, there was no "high value underestimation and low value
overestimation", which was comparable to the accuracy of BESS and GOSIF. However, the simulation accuracy of the other
GPP datasets in Chinaflux was relatively poor, with the $R^2$ of NIRv being only 0.64, while FLUXCOM-ENS, FLUXCOM-
RF, MODIS and Revised EC-LUE were significantly underestimated, with the Sim/Obs being only 0.71-0.89. In the validation
of FLUXNET, the $R^2$ of FLUXCOM-ENS, MODIS, and Revised EC-LUE ranged from 0.57 to 0.67, and the RMSE ranged
from 2.67 to 3.3 gC m$^{-2}$ d$^{-1}$, and exhibited different degrees of underestimation (Figure S9). Other GPP datasets demonstrated
similar performance, with ERF_GPP being the best ($R^2$ = 0.74, RMSE = 2.26 gC m$^{-2}$ d$^{-1}$).

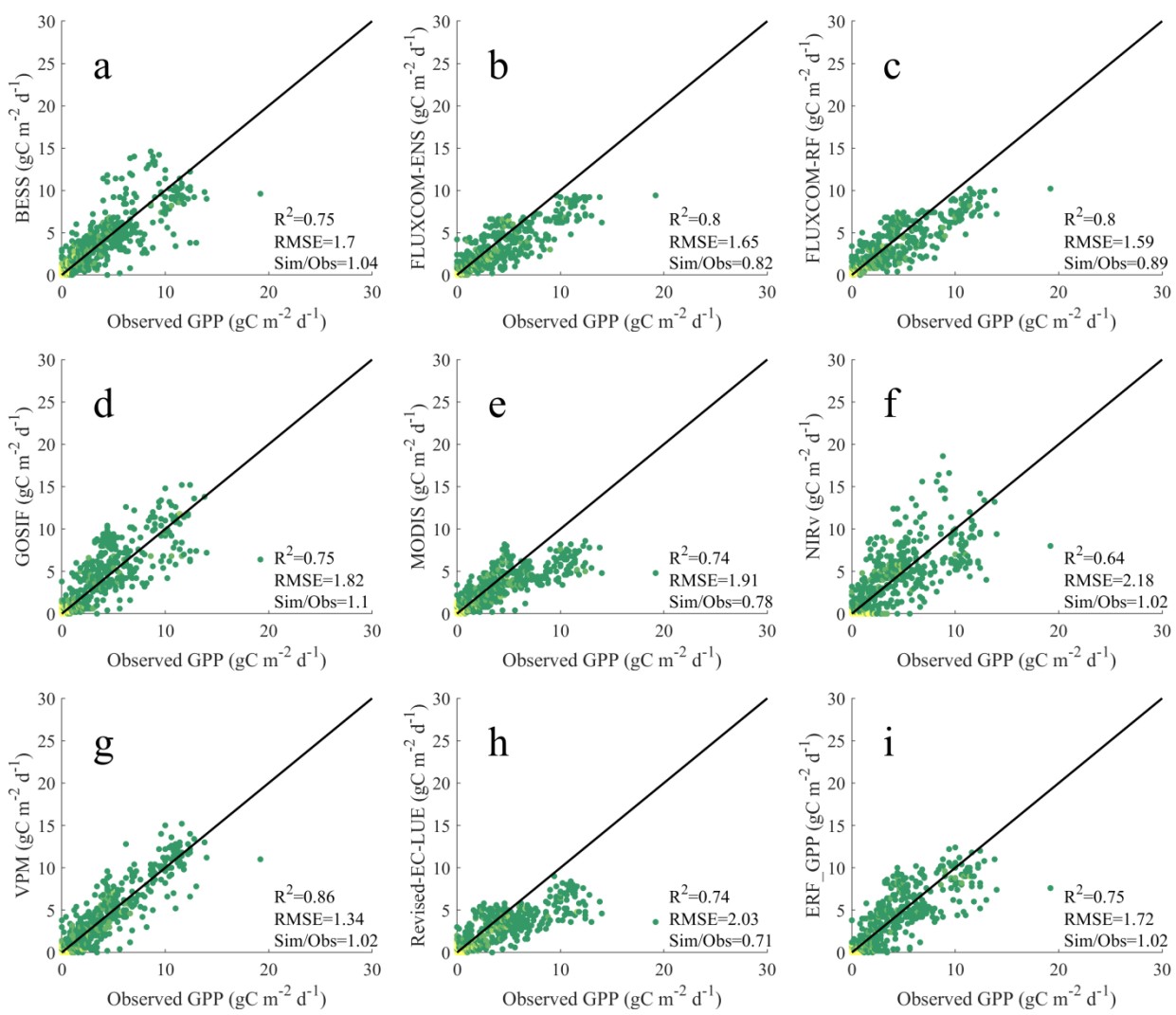

**Figure 6.** Comparison between the GPP datasets and the GPP observations from ChinaFlux. a-i represents BESS, FLUXCOM-ENS, FLUXCOM-RF, GOSIF, MODIS, NIRv, VPM, Revise-EC-LUE, ERF_GPP, respectively.

## 4 Discussion

### 4.1 Performance analysis of different models

After parameter calibration, both LUE and vegetation index models obtained reliable model accuracy. However, noticeable errors persist in different months and subvalues, indicating the prevalent phenomenon of "high value underestimation and low value overestimation" (Figures 1-4). In addition to MODIS, the GPP simulated by the other three LUE models is generally underestimated in winter (Figure 3), which may be caused by biases in the parameters used in meteorological constraints. In

the expression form of the temperature constraint adopted by LUE models, the maximum temperature, minimum temperature and optimum temperature for limiting photosynthesis are all constants, however these values may not be fixed (Huang et al., 2019; Grossiord et al., 2020). A previous study has demonstrated that the GPP estimate could be effectively improved by using dynamic temperature parameters (Chang et al., 2021). Moreover, the form of meteorological constraint is also an important influencing factor. Compared with other LUE models, VPM does not use VPD constraints, but incorporates land surface water index from satellite observations as constraints (Xiao et al., 2004), which may be the reason why the model performs better than other models at high value (Figure 4). Conversely, the two vegetation index models overestimated GPP in winter, and even overestimated by 70% in December. The vegetation index model does not consider meteorological constraints that believe that all environmental impacts on vegetation have been included in the vegetation index (kNDVI, NIRv). However, it is a fact that under high temperature or low radiation, the vegetation index may still maintain the appearance of high photosynthesis (greening), while in fact the GPP is low (Doughty et al., 2021; Yang et al., 2018; Chen et al., 2024). Furthermore, the relationship between these vegetation indices and GPP is not robust, and the vegetation indices based on reflectance may have hysteresis (Wang et al., 2022).

Compared to other GPP models, the ERF model demonstrated better performance ($R^2 = 851$). Since there are no physical constraints, the machine learning model needs to find the relationship between explanatory variables and target variable from a large amount of training data (such as GPP=f (LAI,T,P, etc.)). Therefore, the reliability of the model usually depends on the representativeness of the training data. For example, LAI can explain GPP to a large extent, while complex modeling relationships are still needed from LAI to GPP. The difference between the ERF model and the RF model lies in the explanatory variables. The ERF model uses multiple GPP simulations that are more representative and aligned with the target variable, thus making the GPP simulations more accurate. In other words, the ERF model does not need to take into account the uncertainties of the model structure (such as meteorological constraints) and model parameters (such as maximum light use efficiency), but rather focuses on the uncertainties inherent in the simulated GPP. To further clarify the impact of explanatory variables on the ERF model, we conducted a feature importance analysis (Figure S10). From an average of 200 times, the results of the ERF model did not depend on a single GPP simulation. Even GPP$_{MODIS}$, with the highest relative importance, accounted for no more than 25%, suggesting that the ERF model behaves more like a weighted average of multiple GPP simulations. In addition, it is important to emphasize that the accuracy of the ERF model is still robust even for GPP simulations of original parameters (Figure S4), which means that we can try to use this method to integrate the currently published GPP data sets to obtain a more accurate global GPP estimate.

It is worth noting that in the study of Tian et al. (2023), the ERF model was also used to improve the GPP estimate. Our study extends this work in several ways. Firstly, parameter calibration was carried out in our study so that the final validation results are comparable, that is, differences in model performance are mainly due to the uncertainty of the model structure. Secondly, our study focused on the phenomenon of "high value underestimation and low value overestimation" of GPP model, with results indicating that the ERF model performed well across various vegetation types, months, and subvalues. Finally, we

generated the ERF_GPP dataset and validated it on different observational datasets, further confirming the robustness of the
ERF model in GPP estimate.

## 4.2 Robustness of ERF_GPP

Since the current GPP datasets are generated based on remote sensing and FLUXNET GPP observations, there is a strong
similarity in spatial distribution among all GPP datasets. Therefore, the validation of GPP observations independent of
FLUXNET is crucial. Validation results from GPP observations of ChinaFlux indicated that ERF_GPP exhibited good
generalization in China ($R^2$=0.75), which was slightly lower than the accuracy of 5-fold-cross-validation during modeling,
possibly due to the mismatch between the 0.05 °GPP estimate and the footprint of the flux tower (Chu et al., 2021). In addition,
the validation of FLUXNET further confirms the reliability of ERF_GPP. Overall, this is comparable to or slightly better than
the simulation accuracy of current mainstream GPP datasets. We also observed a clear improvement in the spatial maximum
value of ERF_GPP in some corn growing regions, such as the North American Corn Belt (Figure 5c), which is supported by
previous studies showing that C4 crops have much higher GPP peaks than other vegetation types (Yuan et al., 2015; Chen et
al., 2011).
Due to the increasing trend of drought, the constraining effect of water on vegetation is gradually increasing, and some studies
have reported the decoupling phenomenon of LAI and GPP under some specific conditions (Jiao et al., 2021; Hu et al., 2022).
However, in China and India with significant greening, GPP ontinues to increase in most datasets, and ERF_GPP supports this
view. This phenomenon may be attributed to the low drought pressure on croplands in China and India due to irrigation, which
poses less constraint on GPP (Ambika and Mishra, 2020; Ai et al., 2020). The global estimate of ERF_GPP is 132.7 ±2.8 PgC
yr$^{-1}$, which is close to estimates from most previous studies (Wang et al., 2021; Badgley et al., 2019). A study have suggested
that global GPP may reach 150-175 PgC yr$^{-1}$ (Welp et al., 2011), however, there is no further evidence to support this view.
ERF_GPP exhibited higher uncertainty in tropical regions, similar reports have been made in previously published GPP
datasets (Badgley et al., 2019; Guo et al., 2023). The scarcity of flux observations in these regions (Pastorello et al., 2020),
coupled with the well-known issue of cloud pollution and saturation in remote sensing data in the tropics (Badgley et al., 2019),
exacerbates the uncertainty in GPP estimates for these regions. Therefore, in future studies, on the one hand, more flux
observations in tropical regions are needed, and on the other hand, attempts can be made to combine optical and microwave
data to improve GPP estimate.

## 4.3 Limitations and uncertainties

In this study, we improved GPP estimate based on the ERF model. Nonetheless, there are still some limitations and
uncertainties due to the availability of data and methods. First, C4 crop distribution maps were used in our study to improve
estimates of cropland GPP. However, it is important to note that this dataset only represents the spatial distribution of crops
around the year 2000, which introduce uncertainty into GPP simulations of cropland in a few C3 and C4 alternating areas.
Secondly, the ERF model considers six GPP simulations, and it is not clear whether adding more GPP simulations to the model
can further improve the GPP estimate. Finally, our model did not consider the effect of soil moisture on GPP, and some
previous studies have highlighted the importance of incorporating soil moisture in GPP estimates, especially for dry years
(Stocker et al., 2019; Stocker et al., 2018).
**5 Conclusion**
In this study, we compared the performance of the ERF model with other GPP models at the site scale, especially for the
phenomenon of "high value underestimation and low value overestimation", and further developed the ERF_GPP dataset.
Overall, $GPP_{ERF}$ had higher model accuracy, explaining 85.1% of the monthly GPP variations, and demonstrated reliable
accuracy in different months, vegetation types and subvalues. Over the period from 2001 to 2022, the global estimate of
ERF_GPP was $132.7 \pm 2.8$ PgC yr$^{-1}$, corresponding to an increasing trend of 0.42 PgC yr$^{-2}$. Validation results from ChinaFlux
indicated that ERF_GPP had good generalization. For the current emerging GPP estimate models, the ERF model provides an
alternative method that lead to better model accuracy.
**Data and code availability**
The ERF_GPP for 2001-2022 is available at https://doi.org/10.6084/m9.figshare.24417649 (Chen et al., 2023). The spatial
resolution of ERF_GPP is 0.05° and the temporal resolution is monthly. Code is available from the author upon reasonable
request.
**Author contributions**
X.C. and T.X.C. conceived the scientific ideas and designed this research framework. X.C. compiled the data, conducted
analysis, prepared figures. X.C., T.X.C. and Y.F.C. wrote the manuscript. D.X.L., R.J.G., J.D., and S.J.Z. gave constructive
suggestions for improving the manuscript.
**Acknowledgments**
This study was supported by the National Natural Science Foundation of China (No. 42130506, 42161144003 and 31570464)
and the Postgraduate Research & Practice Innovation Program of Jiangsu Province (No. KYCX23_1322).
**Declaration of interests**
The authors have not disclosed any competing interests.

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
