# Peer review of "2001-2022 global gross primary productivity dataset using an ensemble model based on random forest"

_EGUsphere, 2024_

## Author Comment (AC1)

*In their study the authors created two new datasets of gross primary productivity (GPP), one based on remote sensing and environmental predictors and one an ensemble of four existing GPP models. Both models connect predictors and observed GPP using Random Forests. To test the practicality of their approach, the authors compared their two products and the four existing models to FLUXNET site observations. Additionally, they created a global gridded GPP estimate using the ensemble-based approach and performed an independent evaluation using site observations from FluxChina. Improving estimates of global GPP is indeed an important scientific challenge. However, while the reported model metrics suggest a substantial improvement in particular for their ensemble-based model compared to existing models, I am not convinced of the novelty and whether there is indeed a real improvement. My main concerns are the following:*

*The methodology behind the model evaluation is unclear. It seems that all models "saw" the full FLUXNET data during parameter calibration and then final model evaluation (Fig. 1-4) was computed based on the full dataset? Model evaluation should be done on a separate test dataset. If no separate test dataset existed, the ensemble approach might just have learned the typical GPP values of this site and its fluctuations from the patterns in the four other models. There is an independent evaluation included in the paper which does not suffer from this issue (ChinaFlux), however, only 12 sites are included and other existing models show comparable prediction skills.*

**REPLY:** Thanks for your comments. As you said, all models used FLUXNET data set in parameter calibration, but it should be noted that only 70% samples were selected in our parameter calibration each time, and the average value of 200 calibrated parameters was used as the final parameter, so as to avoid obtaining a parameter applicable to the complete FLUXNET. This is also a common practice for model parameter calibration (Badgley et al. 2019, Zheng et al. 2020). The practice of some previous studies was to use 70% of the sample for calibration and the remaining 30% for validation. But they only do this once, or choose the best many times (e.g. Wang et al. 2020), and it is entirely possible to get an accidental parameter that only applies to this one validation set.

Badgley, G., Anderegg, L. D., Berry, J. A., and Field, C. B.: Terrestrial gross primary production: Using NIRV to scale from site to globe, Global change biology, 25, 3731-3740, 2019.

Zheng, Y., Shen, R., Wang, Y., Li, X., Liu, S., Liang, S., Chen, J. M., Ju, W., Zhang, L., and Yuan, W.: Improved estimate of global gross primary production for reproducing its long-term variation, 1982–2017, Earth System Science Data, 12, 2725-2746, 2020.

Wang, S., Zhang, Y., Ju, W., Qiu, B., and Zhang, Z.: Tracking the seasonal and inter-annual variations of global gross primary production during last four decades using satellite near-infrared reflectance data, Science of the Total Environment, 755, 142569, 2021.

Secondly, the main purpose of our parameter calibration is to reduce the impact of the uncertainty of the model parameters on the validation results. The original parameters of these models were calibrated with only a small number of sites (e.g., 95 sites were

used for Revised EC-LUE and 104 for NIRv). Therefore, when we used the original parameters, the results validated by 170 sites (sorry, The 171 sites in the original text are typographical errors) in this study contain **not only the uncertainty of the model structure, but also the uncertainty of the model parameters.** In the revised version, we explain this in detail:

FLUXNET only provides GPP observation and meteorological data, while LAI, FPAR and surface reflectance are not provided, so only remote sensing data can be used. However, there are many sources of remote sensing data, such as MODIS, AVHRR, etc., so using different remote sensing data to calibrate the same GPP model may produce different model parameters. In addition, the number of sites used to calibrate model parameters is also an important influencing factor for model parameters. The original parameters of these models were calibrated with only a small number of sites (e.g., 95 sites were used for Revised EC-LUE and 104 for NIRv). Therefore, to reduce the impact of the uncertainty of the model parameters on simulation results, we did not use original parameters in the model, but carried out parameter calibration and for GPP models according to different vegetation types.

For the ensemble model, we used "5-fold cross-validation" method, which is the most common method for machine learning validation. That is to say, we divide all samples into 5 parts, select 4 of them for modeling each time, and validate the rest once, so that the cycle is repeated five times to obtain the complete validation result. These validation sets are independent, so the validation results are reliable.

To further dispel your doubts, we used the original parameters of these models for validation and the construction and validation of ensemble model. The author of kNDVI did not provide model parameters, so this model was abandoned. In addition, reviewer 2 suggested that MODIS and VPM be added. Therefore, the validation of 5 GPP models and the ensemble model built based on these 5 GPP models are shown in Figure R1, in the GPP simulation using the original parameters, the performance of these GPP models was significantly decreased, $R^2$ ranged from 0.570 to 0.719, RMSE ranged from 2.29 to 3.81 gC m$^{-2}$ d$^{-1}$, in addition, the phenomenon of "high value underestimation and low value overestimation" was also serious. However, the ensemble model exhibited consistent advantages, with $R^2$ significantly higher than other GPP models (0.856). As just mentioned, these GPP models contain uncertainties in model parameters and model structure, which makes them perform poorly, and the excellence of the ensemble model also proves the reliability of the results of this study. In the revised version, we have added the results of this section:

In order to further prove the robustness of the ERF model, we also used GPP models with original parameters for modeling and validation. As shown in Figure S3, the performance of these GPP models decreased significantly, with $R^2$ ranging from 0.570 to 0.719 and RMSE ranging from 2.29 to 3.81 gC m$^{-2}$ d$^{-1}$. The phenomenon of "high underestimation and low overestimation" was also serious. However, the ERF model showed a consistent advantage, with $R^2$ significantly higher than other GPP models (0.856).

[Figure]

Figure R1. Comparison between the GPP simulations of the six models and the GPP observations. a-f represents GPP$_{EC}$, GPP$_{NIRv}$, GPP$_{REC}$, GPP$_{VPM}$, GPP$_{MODIS}$, GPP$_{ERF}$, respectively. The author of kNDVI did not provide model parameters, so this model was abandoned.

In ChinaFlux's GPP validation, we did not validate these GPP models, but rather the published GPP data set and the results of the ensemble model in this study (at 0.05° grid). This is mainly due to the absence of meteorological data at some sites, which made it impossible for us to obtain the GPP simulation of all models at the site scale (500 m). In the revised version, we explain this in detail.

It should be noted that due to the absence of meteorological data from some sites in Chinaflux, we did not validate all GPP models at the site scale (500 m).

*Even for the evaluation performed on a separate test dataset (i.e. ChinaFlux), I wonder whether the good prediction skill of GPPERF is mostly a result of spatial autocorrelation, i.e. by learning the patterns from the four GPP products RFERF basically finds the correct region and predicts the GPP values of the nearest FLUXNET site?*

**REPLY:** Thanks for your comments. As mentioned above, ChinaFlux's site was not involved in the validation of simulation results of all GPP models, but is used to validation results of other GPP datasets and ensemble model at grid (0.05°). The good performance of GPP$_{ERF}$ is not actually a spatial improvement, nor is it the result of spatial autocorrelation, because these GPP observations are a collection of different sites over the months, that is, high values actually indicate the GPP of the growing season, and low values indicate the non-growing season. Therefore, the improvement here is actually the simulation on the time series, which is to improve the phenomenon of "high value underestimation and low value overestimation" emphasized in this study. As shown in Figure R2 and R3, we show the simulation

results of each model at the two sites. It is obvious that GPP$_{EC}$, GPP$_{REC}$ and GPP$_{MODIS}$ on CN-Qia showed obvious underestimation during the growing season. On CH_Lae, GPP$_{kNDVI}$ and GPP$_{VPM}$ were significantly overestimated. In contrast, at both sites, GPP$_{ERF}$ is more consistent with observations, meaning that the good performance of GPP$_{ERF}$ is due to the correction on the time series (although it is not well corrected at all sites). The performance of each model is different at different sites, mainly because the process concerned by each model (meteorological constraints) is different. For example, NIRv and kNDVI do not use constraints in the modeling process, while other models add some constraints such as temperature. In the revised version, we have added the results of this section:

Further presentations were made at two typical sites, it was obvious that GPP$_{EC}$, GPP$_{REC}$ and GPP$_{MODIS}$ on CN-Qia showed obvious underestimation during the growing season (Figure S4). On CH_Lae, GPP$_{kNDVI}$ and GPP$_{VPM}$ were significantly overestimated (Figure S5). In contrast, at both sites, GPP$_{ERF}$ was more consistent with observations, meaning that the good performance of GPP$_{ERF}$ was due to the correction on the time series (although it was not well corrected at all sites).

[Figure]

Figure R2. Performance of each GPP model on CN-Qia.

[Figure]

Figure R3. The performance of each GPP model on CH_Lae.

*The authors' remote sensing and environmental predictors model seems to be similar to the FLUXCOM approach. I wonder what is the advantage and why FLUXCOM is not included in the comparison?*

**REPLY:** Thanks for your comments. As you said, the RF model we used is actually one of the GPP estimation models in FLUXCOM. The purpose of using this model is to compare it with the ensemble model, nothing more, because both use the random forest method, then the difference in validation results is mainly due to the influence of the input data set (GPP$_{RF}$: remote sensing and environmental variables; GPP$_{ERF}$: GPP simulation). We did not use the FLUXCOM dataset when comparing the results of the ensemble model with other GPP datasets, mainly because it did not provide data with 0.05° resolution, and only one set of 0.5° data fit the Chinaflux validation set time range (2001-2018). Based on your comments, we have added a comparison of FLUXCOM in the revised version. We also added validation of the VPM and MODIS GPP datasets. As shown in Figure R4, in Chinaflux's validation, the accuracy of FLUXCOM is reliable, but it shows a certain underestimation. In contrast, VPM showed better performance, which may be related to their preprocessing of the input data. In the revised version, we have added a description of the relevant content in the results section:

As shown in Figure 6, ERF_GPP and other GPP datasets were validated using GPP observations from ChinaFlux. Of all the models, $GPP_{VPM}$ has the best performance, with $R^2$ of 0.86 and RMSE of gC m$^{-2}$ d$^{-1}$. ERF_GPP also had a high generalization, $R^2$ of 0.75, RMSE of 1.72 gC m$^{-2}$ d$^{-1}$, there was no "high value underestimation and low value overestimation", which was comparable to the simulation accuracy of BESS and GOSIF. However, the simulation accuracy of the other GPP datasets in Chinaflux was relatively poor, with the $R^2$ of NIRv being only 0.64, while FLUXCOM, MODIS and Revised EC-LUE was significantly underestimated, with the Sim/Obs being only 0.71-0.82.

[Figure]

Figure R4. Comparison between the GPP datasets and the GPP observations from ChinaFlux. a-h represents BESS, FLUXCOM, GOSIF, MODIS, NIRv, VPM, Revise-EC-LUE, ERF_GPP, respectively.

*The authors recalibrated the parameters underlying the four existing models but the justification for this action is unclear. I would like to see a comparison with the original models to see whether this indeed led to improvements in model performance.* **REPLY:** Thanks for your comments. As stated in the first point, the calibration parameters are used to compare the performance differences between models considering only the uncertainty of the model structure. For GPP simulation of original parameters, the ensemble model also showed superior performance (Figure R1).

*Several existing GPP datasets are only shown in the comparison to ChinaFlux but were not included in the ensemble-based product. Vice versa, two of the models used in the FLUXNET comparison were omitted from the ChinaFlux comparison. I wonder why the authors selected these four models (EC-LUE, Revised-EC-LUE, GPP-kNDVI, GPP-NIRv) in the ensemble approach even though the comparison in Fig. 6 suggests other products perform much better? If the reason is the spatial resolution this should be better explained.*

**REPLY:** Thanks for your comments. In the first point, we explain why GPP models and ensemble model are not validated using ChinaFLUX. In response to your comments, we have added the results of validation of GPP datasets and ensemble models using FLUXNET data in the revised version. Similarly, we extracted 0.05° MODIS land use covering the flux tower and used the site for analysis when the vegetation types of the flux tower were consistent with MODIS land use. In the end, 52 sites from FLUXNET were used. As shown in Figure R5, the validation results of the ensemble model are significantly better than those of other GPP datasets. However, underestimation is shown in the high value, which may be due to the inconsistency between the 0.05° coarse resolution and the flux tower footprint. In the revised version, we have added a description of the relevant content in the results section:

In the validation of FLUXNET, the $R^2$ of FLUXCOM, MODIS, and Revised EC-LUE ranged from 0.57 to 0.67, and the RMSE ranged from 2.67 to 3.3 gC m$^{-2}$ d$^{-1}$, and showed different degrees of underestimation (Figure S8). Other GPP datasets showed similar performance, with ERF_GPP being the best ($R^2$=0.74, RMSE = 2.26 gC m$^{-2}$ d$^{-1}$). Notably, in the high values, all models exhibited significant underestimation, which may be caused by the 0.05° resolution being inconsistent with the flux tower footprint.

[Figure]

Figure R5. Comparison between the GPP datasets and the GPP observations from FLUXNET. a-h represents BESS, FLUXCOM, GOSIF, MODIS, NIRv, VPM, Revise-EC-LUE, ERF_GPP, respectively.

For the four (now is six) GPP models selected in the ensemble model, this is justified and sorry not to be mentioned in the original article. The GPP models mainly include process model, light use efficiency model, vegetation index model and machine learning model. The process model is very complex, many parameters are considered, and the accuracy of the models is not very outstanding, although they are more suitable for the process of photosynthesis. We expect the ensemble model to improve the performance of the model without being too complex, so we mainly chose a few representative models that are widely used. In the revised version, we explain this in detail. At the same time, at the suggestion of reviewer 2, we also added VPM and MODIS in the revised version. In other words, there are 6 GPP models in the ensemble model in the revised version.

In this study, six independent models were selected to estimate GPP. These models are widely used with few model parameters and have shown reliable model accuracy in previous studies.

In addition, we added a section on the effect of the amount of GPP on the accuracy of the ensemble model. As shown in Table R1, as the number of GPP in the ensemble model increases, the model performance gains gradually decrease.

Table R1. Effect of the GPP number in the ERF model on model performance

| GPP number | 2 | 3 | 4 | 5 |
|---|---|---|---|---|
| $R^2$ | $0.793\pm0.024$ | $0.824\pm0.011$ | $0.836\pm0.004$ | $0.845\pm0.001$ |
| RMSE | $1.798\pm0.104$ | $1.658\pm0.052$ | $1.600\pm0.022$ | $1.556\pm0.009$ |
| Sim/Obs | $1\pm0.001$ | $0.999\pm0.000$ | $1\pm0.000$ | $1\pm0.000$ |

*Minor comments:*

*L18: Remove "a".*

**REPLY:** Thanks for your comments. We have corrected this error in the revised version.

*L33: I think you mean "to the terrestrial carbon cycle".*

**REPLY:** Thanks for your comments. This sentence has since been modified to Gross primary productivity (GPP) is the largest carbon flux in the global carbon cycle, and it is also the input of carbon into the terrestrial carbon cycle.

*L38: Unclear, is this about remote sensing-based estimates or GPP estimates in general? Also it is unclear how the approach applied in this study helps with the problems mentioned in the following sentences. Overall the introduction lacks connectivity.*

**REPLY:** Thanks for your comments. This refers to the models that involve remote sensing data in the estimation of GPP.

In this paragraph, we want to emphasize the problems existing in these GPP models. Of course, many of the problems mentioned have not been solved in our research. Therefore, we have sorted out this paragraph again, focused on the uncertainty of several GPP models, and introduced the ensemble model.

The light use efficiency (LUE) model is one of the most widely used models for estimating GPP. It assumes that GPP is proportional to the photosynthetically active radiation absorbed by vegetation, and optimizes the spatio-temporal pattern of GPP through meteorological constraints such as temperature and water (Pei et al., 2022). However, the form of these meteorological constraints varies greatly, and this difference alone can lead to a difference of more than 10% in the explanatory power of the models (Yuan et al., 2014). Recent studies have proposed some new vegetation indices that have been shown to be effective proxies for GPP through theoretical derivation and validation by observations (Badgley et al., 2017; Camps-Valls et al., 2021). However, these vegetation indices often use only remote sensing data as an input for estimating long-term GPP without considering meteorological factors, which has led to some controversy (Chen et al., 2024; Dechant et al., 2020). Both the LUE model and the vegetation index model use a combination of linear mathematical formulas to estimate GPP. However, ecosystems are highly complex and the biases introduced into a process by this numerical model increase the uncertainty in the estimates of the final product (GPP). The machine learning model has shown in previous studies that it has great potential to improve GPP estimates (Jung et al., 2020). This model is trained by non-physical means directly using GPP observations and selected environmental and vegetation variables, and the performance of the model depends on the number and quality of the observed data and the

representativeness of the input data. Machine learning has also been widely used in recent years due to its advantages such as the fact that no parameter calibration is required and the reliable model accuracy. Nevertheless, direct validation from flux tower of FLUXNET shows that the model typically explains only about 70% of the monthly variations in GPP, with similar performance to other models (Wang et al., 2021; Badgley et al., 2019; Zheng et al., 2020; Jung et al., 2020). Due to the deviation of the model structure, there is a common problem in these models, that is, the estimation of monthly extreme GPP is poor, and the phenomenon of "high value overestimation and low value overestimation" occurs (Zheng et al., 2020). Especially for extremely high values, which usually occur during the growing season and largely determine the annual value and interannual fluctuations of GPP, this underestimation may hinder our understanding of the entire carbon cycle.

*L48: Unclear, do you mean the models assume a positive relationship between CO2 and GPP while it is actually negative? Or that CO2 fertilization started to saturate?*

**REPLY:** Thanks for your comments. As you said, it means that the effect of $CO_2$ fertilization tends to be saturated, that is, the positive impact of $CO_2$ fertilization on GPP is weakening. Considering that the ensemble model in this study also did not include this saturation $CO_2$ fertilization effect, we deleted this sentence to avoid misunderstanding.

*L55: Is this for the same region?*

**REPLY:** Thanks for your comments. This is true in some areas where C3 and C4 are grown alternately. This sentence has been deleted due to changes in the introduction.

*L73: "low"?*

**REPLY:** Thanks for your comments. We have corrected this error in the revised version.

*L85: "ERA". Also references are missing.*

**REPLY:** Thanks for your comments. We have corrected this error and added references in the revised version.

*L108: How were they resampled?*

**REPLY:** Thanks for your comments. For higher resolution data, we gridded the dataset to 0.05° by averaging all pixels whose center fell within each 0.05° grid cell for upscaling. For lower resolution data, we used the nearest neighbor resampling to 0.05°. In the revised version, we explained the resampling method in detail.

Finally, for higher resolution data, we gridded the dataset to 0.05° by averaging all pixels whose center fell within each 0.05° grid cell for upscaling. For lower resolution data, we used the nearest neighbor resampling to 0.05°. In addition, MODIS data were aggregated to a monthly scale to ensure spatio-temporal consistency.

*L115: Why only 171 sites? Did the other sites not contain any high-quality years?*

**REPLY:** Thanks for your comments. As you said, there are some sites that only have one or two years of data, so it's not unusual to have years without quality data. For example, AU-Lox only has data for 2008-2009, and US-Wi1 only has data for 2003. Therefore, based on the quality screening criteria, only 170 sites were used in the end (sorry, The 171 sites in the original text are typographical errors).

*L120: The paper often mentions "remote sensing models" but the atmospheric data is actually from a reanalysis (ERA5) or FLUXNET.*

**REPLY:** Thanks for your comments. There is also a class of models that estimate GPP driven only by climate and land use data, known as dynamic global vegetation models. These models do not involve remote sensing data in the estimation of GPP, so they are fundamentally different from the models mentioned in this study. In the revised edition, we call these models "GPP models", "LUE models," and "Vegetation Index models".

*L121: What is "traditional random forest model"? The authors often mix the nature of the data (e.g. remote sensing) and modelling approach (e.g. random forests).*

**REPLY:** Thanks for your comments. The traditional random forest model refers to the model using remote sensing data and environmental factors in previous studies. In the revised version, we redefine this concept. In addition, we define these models uniformly as GPP models under different methods.

*L125: Table 1 says EC-LUE also considers CO2.*

**REPLY:** Thanks for your comments. This is a mistake, as the Revised EC-LUE model simply divides the leaves into sunlit and shaded leaves. In the revised version, we have corrected this error.

*L127: SIF was not mentioned previously.*

**REPLY:** Thanks for your comments. The SIF here is sun-induced chlorophyll fluorescence, and in the revised version, we have corrected this error.

*L129 A brief summary of random forests is needed. Also why did you choose these four predictors? I assume adding more variables would increase model performance.*

**REPLY:** Thanks for your comments. We briefly introduce random forest methods in the revised version.

Random forest is an ensemble learning algorithm that combines the outputs of multiple decision trees to produce a single result, and is commonly used for classification and regression problems. In the regression problem, the output result of each decision tree is a continuous value, and the average of the output results of all decision trees is taken as the final result.

Following your suggestions, we adjusted the input data in the random forest model, including LAI, FPAR, T, TMIN, VPD, DifSR and DirSR, a total of 7 variables. The addition of $CO_2$ does not make sense because it does not characterize the effect of $CO_2$ fertilization. In addition, NIRv and kNDVI are not included in the model because these two inputs are proxies for GPP and are converted to GPP using only a linear equation. If these two variables are included, the model is essentially the same as the ensemble model. To further dispel your doubts, we present the results of models incorporating NIRv and kNDVI, but to avoid repetitive results, this part is not presented in the paper.

As shown in Figure R6-R9, the $R^2$ of the random forest model using 7 variables is 0.815. Although it is slightly better than other GPP models, it still lags behind the ensemble model. In addition, the performance of the model in different months, different vegetation types and different subvalues is also worse than that of the ensemble model. In other words, the result is similar to the original paper.

[Figure]

Figure R6. Comparison between the GPP simulations of the eight models and the GPP observations. a-h represents GPP$_{EC}$, GPP$_{NIRv}$, GPP$_{kNDVI}$, GPP$_{REC}$, GPP$_{VPM}$, GPP$_{MODIS}$, GPP$_{RF}$, GPP$_{ERF}$, respectively.

[Figure]

Figure R7. Performance of the eight models in each month. a, b and c represent $R^2$, RMSE, and Sim/Obs respectively.

[Figure]

Figure R8. The performance of the eight models on different vegetation types. a, b and c represent $R^2$, RMSE, and Sim/Obs respectively.

[Figure]

Figure R9. Performance of eight models in different subvalues.

As shown in Figure R10, R$^2$ of the random forest model using 9 variables is 0.845, which is similar to the performance of the ensemble model, as mentioned earlier, the two models are essentially the same. However, in terms of vegetation type (underestimation of C4 crops, overestimation of SHR and WET), and subvalues (underestimation of high value), the performance of the model also remained gap with that of the ensemble model.

[Figure]

Figure R10. Performance of the random forest model using 9 variables.

*L132: "multi-model".*

**REPLY:** Thanks for your comments. We have corrected this error in the revised version.

*L137: Provide information about data source. If I understood correctly, e.g. FPAR is from MODIS (500m) while AT from FLUXNET? And ERA5 AT is only used for the global prediction? This is confusing. Also where is the NIR data from?*

**REPLY:** Thanks for your comments. In the revised version, we further clarified the source of the data. FLUXNET not only provided GPP observations, it also provided meteorological data, and ERA5-land was used for global GPP estimates. In addition, the red-band and near-infrared data were also from MODIS.

GPP observations from the FLUXNET 2015 dataset, which includes carbon fluxes and meteorological variables from more than 200 flux sites around the world (Pastorello et al., 2020). GPP cannot be obtained directly from the flux site and usually needs to be obtained by dismantling the Net Ecosystem Exchange. We chose a monthly level GPP based on the nighttime partitioning method and retained only high quality data (NEE_VUT_REF_QC > 0.8) for every year, and finally selected 170 sites with 10932 monthly values for this study. In addition, average air temperature, total solar radiation and VPD on the monthly scale were selected.

Since part of the data required for the model is not directly available at the flux site, LAI and FPAR were extracted from MOD15A2H (500 m), surface reflectance (red band, near infrared band, blue band and shortwave infrared band) are derived from MCD43A4 (500m) and MOD09A1 (500m).

*L140: What differences do you mean?*

**REPLY:** Thanks for your comments. FLUXNET only provides GPP observation and meteorological data, while LAI, FPAR and other data are not provided, so only remote sensing data can be used. However, there are many sources of remote sensing data, such as MODIS, AVHRR, etc., so using different remote sensing data to calibrate the same GPP model may produce different model parameters. In addition, the number of sites used to calibrate model parameters is also an important influencing factor for model parameters. Therefore, in the revised version, this sentence has been modified to

FLUXNET only provides GPP observation and meteorological data, while LAI, FPAR and surface reflectance are not provided, so only remote sensing data can be used. However, there are many sources of remote sensing data, such as MODIS, AVHRR, etc., so using different remote sensing data to calibrate the same GPP model may produce different model parameters.

*L155: The model overestimates or underestimates.*

**REPLY:** Thanks for your comments. We have corrected this error in the revised version.

*F160: How many? Again, references are missing.*

**REPLY:** Thanks for your comments. The flux observations provided by Chinaflux are not consolidated in a single article, so it is still being updated, and it is difficult to specify how many sites are available. For references, we show them in Table S1 because every site has one reference.

*L166: Lack of consistency, GPPERF, ERF_GPP or "random forest-based ensemble model"? Or does GPPERF refer to the site predictions while ERF_GPP to the global ones? Again, why are some models thrown out in this step while others are included for the first time?*

**REPLY:** Thanks for your comments. $GPP_{ERF}$ represents the site simulation and ERF_GPP represents the global GPP. Random forest-based ensemble model represents GPP simulation method. In the revised version, we define these.

In this step, we aim to compare ERF_GPP with some of the GPP datasets that are widely used, including GPP datasets generated by other models because these datasets are generated by other methods, such as BESS, which is based on process models, and GOSIF, which is GPP generated by Sun-induced chlorophyll fluorescence. The models used in this study are not all compared in this step, because not all models have relevant data sets, such as kNDVI.

*L185: What do you mean by changes in cropland? Do you mean seasonal changes in cropland GPP?*

**REPLY:** Thanks for your comments. As you said, this refers to the seasonal variation of GPP in cropland. In the revised version, this sentence has been modified to

It is worth noting that compared to other vegetation types, the RMSE was highest for cropland, with 6 out of 8 models for C4 crop exceeding 3 gC m$^{-2}$ d$^{-1}$, suggesting that these existing GPP models may not properly capture the seasonal changes in cropland GPP.

*Fig. 2+Fig. S3 Why are the metrics different? Is Fig. S3 the mean of the individual sites while Fig. 2 the mean of all data?*

**REPLY:** Thanks for your comments. As you said, Fig. S3 is the mean of the individual sites, and Fig.2 is all the data. We found that the mean of the individual sites was not very reasonable, which was deleted in the revised version.

*L207: "models". This error occurs several times in the manuscript.*

**REPLY:** Thanks for your comments. We have corrected this error in the revised version.

*L215: What do you mean by extreme? The highest values (>10 gC/m2/d)? Does this represent 33% of all data?*

**REPLY:** Thanks for your comments. This extreme is actually more of an empirical distinction, such as the high value (>15 gC m$^{-2}$ d$^{-1}$, redefined in the revised version), which means that the GPP for that month >450 gC m$^{-2}$ month$^{-1}$, no doubt only some sites can achieve the extreme high value. In addition, it can also be found in Figure 2 that some sites have a significant underestimation in the high value, which is also one of the criteria for empirical discrimination. The use of percentages also requires an empirical discrimination, as there is no precedent for validating GPP in extreme.

*Fig. S2: Why is there an extra panel for site 1? Why don't you also show the FLUXNET sites?*

**REPLY:** Thanks for your comments. In the revised version, we show all GPP observation sites in Fig.S2.

*In general, having a native English speaker review the text would enhance its quality.*

**REPLY:** Thanks for your comments. In the revised version, we have made corrections to the language section.

---

## Author Comment (AC2)

*This study offers a contribution to global gross primary production (GPP) mapping, developing an ensemble model based on random forest algorithm. This model inputs GPP estimations from various remote sensing-based models, showing superior accuracy by explaining 83.7% of GPP variations across 171 sites, outperforming traditional models. It estimates the global GPP to be 131.2 PgC yr-1 from 2001-2022, with an increasing trend. While the authors have done a lot of work and the work is significant, the paper could benefit from a more comprehensive consideration of certain details and improvements in writing clarity.*

*In Section 2.3, the authors selected specific models as input variables for the ERF model. However, other widely applied models such as the P model, VPM model, MODIS GPP algorithm, and NIRvP for vegetation indices have not been considered. What was the rationale behind selecting these four models? Furthermore, in comparing global results, why were certain products chosen, such as VPM, MODIS, and FLUXCOM data, especially considering FLUXCOM also employs machine learning methods and has released a new version of its data (FLUXCOMX)? Additionally, it appears the ECGC has only recently been launched and may not be as "widely used" as mentioned in the manuscript.*

**REPLY:** Thanks for your comments. For the four (now is six) GPP models selected in the ensemble model, this is justified and sorry not to be mentioned in the original article. The GPP models mainly include process model, light use efficiency model, vegetation index model and machine learning model. The process model is very complex, many parameters are considered, and the accuracy of the models is not very outstanding, although they are more suitable for the process of photosynthesis. We expect the ensemble model to improve the performance of the model without being too complex, so we mainly chose a few representative models that are widely used. In the revised version, we explain this in detail.

In this study, six independent models were selected to estimate GPP. These models are widely used with few model parameters and have shown reliable model accuracy in previous studies.

At the same time, according to your suggestion, we have also added VPM and MODIS in the revised version. In other words, there are 6 GPP models in the ensemble model in the latest version. As shown in Figure R1-R4, the result is similar to the original paper. In all respects, the performance of the ensemble model is best.

[Figure]

Figure R1. Comparison between the GPP simulations of the eight models and the GPP observations. a-h represents $GPP_{EC}$, $GPP_{NIRv}$, $GPP_{kNDVI}$, $GPP_{REC}$, $GPP_{VPM}$, $GPP_{MODIS}$, $GPP_{RF}$, $GPP_{ERF}$, respectively.

[Figure]

Figure R2. Performance of the eight models in each month. a, b and c represent $R^2$, RMSE, and Sim/Obs respectively.

[Figure]

Figure R3. The performance of the eight models on different vegetation types. a, b and c represent $R^2$, RMSE, and Sim/Obs respectively.

[Figure]

Figure R4. Performance of eight models in different subvalues.

We didn't consider the P model and NIRvP. For the P model, although it is the structure of the LUE model, the calculation of the Photo respiratory compensation point parameter of this model is actually very complicated, which is similar to the process model. This point violates the basic criteria for selecting GPP models in this study. For NIRvP, in a recent study, we found that the model underestimated the impact of drought on GPP by not taking into account environmental constraints (Chen et al, 2024). That is, in dry years, the negative anomaly of GPP is very small, which is obviously inconsistent with the observation. Due to this shortcoming, we do not consider using this model to estimate the global GPP, although its performance may be similar to other models. In addition, we added a section on the effect of the amount of GPP on the accuracy of the ensemble model. As shown in Table R1, as the number of GPP in the ensemble model increases, the model performance gains gradually decrease.

Table R1. Effect of the GPP number in the ERF model on model performance

| GPP number | 2 | 3 | 4 | 5 |
|---|---|---|---|---|
| $R^2$ | $0.793 \pm 0.024$ | $0.824 \pm 0.011$ | $0.836 \pm 0.004$ | $0.845 \pm 0.001$ |
| RMSE | $1.798 \pm 0.104$ | $1.658 \pm 0.052$ | $1.600 \pm 0.022$ | $1.556 \pm 0.009$ |
| Sim/Obs | $1 \pm 0.001$ | $0.999 \pm 0.000$ | $1 \pm 0.000$ | $1 \pm 0.000$ |

Chen, X., Chen, T., Liu, S., Chai, Y., Guo, R., Dai, J., ... & Wei, X. (2024). Vegetation Index-Based Models Without Meteorological Constraints Underestimate the Impact of Drought on Gross Primary Productivity. Journal of Geophysical Research: Biogeosciences, 129(1), e2023JG007499.

*The authors compare the ERF model with a traditional random forest (RF) model. Table 2 indicates that the traditional RF model used only 4 variables, while the ERF model incorporates several GPP estimation models. However, it actually includes even more variables, such as kNDVI, NIRv, FPAR, CO2, dif/dir SR, etc. The ERF model contains more variables than the RF model, but for a fair comparison, the same data should be used. Would the accuracy of the ERF model still surpass that of the RF model if an RF model were constructed using all data inputs from the ERF model?*

**REPLY:** Thanks for your comments. Following your suggestions, we adjusted the input data in the random forest model, including LAI, FPAR, T, TMIN, VPD, DifSR and DirSR, a total of 7 variables. The addition of $CO_2$ does not make sense because it does not characterize the effect of $CO_2$ fertilization. In addition, NIRv and kNDVI are not included in the model because these two inputs are proxies for GPP and are converted to GPP using only a linear equation. If these two variables are included, the model is essentially the same as the ensemble model. To further dispel your doubts, we present the results of models incorporating NIRv and kNDVI, but to avoid repetitive results, this part is not presented in the paper.

As shown in Figure R1-R4, the $R^2$ of the random forest model using 7 variables is 0.815. Although it is slightly better than other GPP models, it still lags behind the ensemble model. In addition, the performance of the model in different months, different vegetation types and different subvalues is also worse than that of the ensemble model. In other words, the result is similar to the original paper.

As shown in Figure R5, $R^2$ of the random forest model using 9 variables is 0.845, which is similar to the performance of the ensemble model, as mentioned earlier, the two models are essentially the same. However, in terms of vegetation type (underestimation of C4 crops, overestimation of SHR and WET), and subvalues (underestimation of high value), the performance of the model also remained gap with that of the ensemble model.

[Figure]

Figure R5. Performance of the random forest model using 9 variables.

*Why did the authors opt to estimate monthly GPP instead of daily? Are the estimation results from different models in the ERF model aggregated from daily to monthly, or are they directly estimating monthly GPP? If monthly, how are parameters like Solar Zenith Angle adjusted when optimizing the rECLUE model?*

**REPLY:** Thanks for your comments. All the results of the model simulation were carried out on the monthly scale. If it is a daily scale GPP simulation, even at 0.05 resolution, it will take a lot of time, so we did not do daily scale GPP simulation. For the solar zenith angle parameter in Revised-EC-LUE, we use the solar zenith angle in the middle of each month as the solar zenith angle of the current month, which is a simplification. Compared with several important parameters that affect GPP simulation, the effect of this parameter is negligible.

*In Table 2, the EC-LUE model considers VPD and CO2, which the original model does not. The supplementary documents indicate that the authors modified the EC-LUE model, thus it is no longer the original EC-LUE model. The only difference between it and the rECLUE model seems to be the consideration of sunlit and shaded leaves. Given that Figure 1 shows minimal differences between them, does including it as an input for the ERF model result in redundancy with rECLUE?*

**REPLY:** Thanks for your comments. First of all, we did not modify the EC-LUE, we used the version published by Yuan et al (2019). As you said, based on our results, the difference between the two models is really not obvious. However, we wish to retain this result because a secondary purpose of our study was to compare the performance differences of these models after parameter calibration.

To address your concerns, our study adds an additional analysis of using different numbers of GPP models in the ensemble model to further compare the performance differences in the final results. As shown in Table R1, As the number of GPP in the ensemble model increases, the model performance gains gradually decrease.

Yuan, W., Zheng, Y., Piao, S., Ciais, P., Lombardozzi, D., Wang, Y., ... & Yang, S. (2019). Increased atmospheric vapor pressure deficit reduces global vegetation growth. Science advances, 5(8), eaax1396.

*The introduction requires careful revision as many uncertainties or current issues listed by the authors seem not to be addressed in this manuscript.*

**REPLY:** Thanks for your comments. The revised introduction highlights the uncertainties of several GPP models and introduces ensemble model.

The light use efficiency (LUE) model is one of the most widely used models for estimating GPP. It assumes that GPP is proportional to the photosynthetically active radiation absorbed by vegetation, and optimizes the spatio-temporal pattern of GPP through meteorological constraints such as temperature and water (Pei et al., 2022). However, the form of these meteorological constraints varies greatly, and this difference alone can lead to a difference of more than 10% in the explanatory power of the models (Yuan et al., 2014). Recent studies have proposed some new vegetation indices that have been shown to be effective proxies for GPP through theoretical derivation and validation by observations (Badgley et al., 2017; Camps-Valls et al., 2021). However, these vegetation indices often use only remote sensing data as an input for estimating long-term GPP without considering meteorological factors, which has led to some controversy (Chen et al., 2024; Dechant et al., 2020). Both the LUE model and the vegetation index model use a combination of linear mathematical formulas to estimate GPP. However, ecosystems are highly complex and the biases introduced into a process by this numerical model increase the uncertainty in the estimates of the final product (GPP). The machine learning model has shown in previous studies that it has great potential to improve GPP estimates (Jung et al., 2020). This model is trained by non-physical means directly using GPP observations and selected environmental and vegetation variables, and the performance of the model depends on the number and quality of the observed data and the representativeness of the input data. Machine learning has also been widely used in recent years due to its advantages such as the fact that no parameter calibration is required and the reliable model accuracy. Nevertheless, direct validation from flux tower of FLUXNET shows that the model typically explains only about 70% of the monthly variations in GPP, with similar performance to other models (Wang et al., 2021; Badgley et al., 2019; Zheng et al., 2020; Jung et al., 2020). Due to the deviation of the model structure, there is a common problem in these models, that is, the

estimation of monthly extreme GPP is poor, and the phenomenon of "high value overestimation and low value overestimation" occurs (Zheng et al., 2020). Especially for extremely high values, which usually occur during the growing season and largely determine the annual value and interannual fluctuations of GPP, this underestimation may hinder our understanding of the entire carbon cycle.

*Some detailed comments:*

*L41: The authors suggest poor estimation accuracy partly because remote sensing models cannot fully represent photosynthesis. Does the ERF model overcome this limitation?*

**REPLY:** Thanks for your comments. The ERF model also does not fully address this problem, but only improves the estimation of the GPP. In the revised version, this sentence has been deleted.

*L46-47: What does "this process may be missing" refer to? Is it the CO2 fertilization effect or a negative trend influenced by CO2? If it's the fertilization effect, many models already consider its impact. If it refers to a negative trend, what improvements have been made in the ERF model? I think this negative trend might not be incorporated into the model.*

**REPLY:** Thanks for your comments. As you said, it means that the effect of $CO_2$ fertilization tends to be saturated, that is, the positive impact of $CO_2$ fertilization on GPP is weakening. Considering that the ensemble model in this study also did not include this saturation $CO_2$ fertilization effect, we deleted this sentence to avoid misunderstanding.

*L52: The authors note significant differences in the same vegetation types across different regions, but it seems the ERF model did not address this variability when optimizing parameters and developing the model.*

**REPLY:** Thanks for your comments. We agree with you that this sentence has been deleted.

*L54-55: It's unclear what this typical example refers to. Parameters for C3 and C4 vegetation inherently need to be considered separately, representing two different vegetation types.*

**REPLY:** Thanks for your comments. Although C3 and C4 are two types of planting, C3 and C4 crops were not divided in many previous studies. Here we want to emphasize the difference between C3 and C4 in the growing season, in the revised version, this sentence has been deleted.

*L56-60: Environmental factors add to GPP estimation uncertainty. How have the authors improved or reduced this uncertainty, given that most models already account for environmental factors?*

**REPLY:** Thanks for your comments. In the ERF model, the uncertainty of these environmental constraints has actually been propagated into the simulated GPP, that is, during the modeling process, the model only needs to consider the uncertainty of the simulated GPP. Accordingly, for other GPP models, there is still the influence of the uncertainty of environmental constraints. In the discussion section of the revised version, we have added an explanation of the relevant content.

In other words, the ERF model does not need to take into account the uncertainties of the model structure (such as meteorological constraints) and model parameters (such as maximum light use efficiency), but only the uncertainties of the simulated GPP.

*L69-70: Tian et al. (2023) also applied ML models to multi-model ensembles. What are the innovative aspects of this study compared to their research?*

**REPLY:** Thanks for your comments. Compared with Tian et al. (2023), our study is a further extension of applying an ensemble model to GPP estimation. There is a big difference compared to their study. Firstly, parameter calibration was carried out in our study so that the final validation results were comparable, that is, the difference in model performance was mainly due to the uncertainty of the model structure. Secondly, our research focuses on the phenomenon of "low value overestimation and high value underestimation" of the GPP model, and the research results show that the ensemble model has a good performance in different vegetation types, different months, and different subvalues. Finally, the ERF model was used to estimate the global GPP and validated on different observational data sets, which further proves the robustness of the ERF model in GPP estimation. In the discussion section of the revised version, We explained the differences between the results of this study and theirs.

It is worth noting that in the study of Tian et al. (2023), the ERF model was also used to improve the GPP. On this basis, our research is further extended. Firstly, parameter calibration was carried out in our study so that the final validation results were comparable, that is, the difference in model performance was mainly due to the uncertainty of the model structure. Secondly, our research focuses on the phenomenon of "low value overestimation and high value underestimation" of the GPP model, and the results show that the ERF model had a good performance in different vegetation types, different months, and different subvalues. Finally, the ERF model was used to estimate the global GPP and validated on different observational data sets, which further proves the robustness of the ERF model in GPP estimation.

*L85: How is ERA5-LAND data procesed in coastal regions? What is the reason for choosing temperature and radiation data from ERA5-Land and ERA5 respectively (this distinction should be made clear in Table 1)?*

**REPLY:** Thanks for your comments. For coarser data conversions to 0.05°, we used the nearest neighbor resampling method. We do the same in the coastal areas. There is no direct radiation in ERA-land, so we used ERA5 monthly data on single levels. In the revised version, we illustrate this in Table1.

Finally, for higher resolution data, we gridded the dataset to 0.05° by averaging all pixels whose center fell within each 0.05° grid cell for upscaling. For lower resolution data, we used the nearest neighbor resampling to 0.05°. In addition, MODIS data were aggregated to a monthly scale to ensure spatio-temporal consistency.

*L104: What does "reference year" mean? How are different datasets aggregated to 0.05 degrees?*

**REPLY:** Thanks for your comments. In the process of calculating the global GPP, land use data is needed. For 2001-2022, we all use data from the same year (i.e., reference year). The simulation was conducted at a resolution of 0.05°, so the effect of

land use change on GPP can be negligible. For higher resolution data, we gridded the dataset to 0.05° by averaging all pixels whose center fell within each 0.05° grid cell for upscaling. For lower resolution data, we used the nearest neighbor resampling to 0.05°. In the revised version, we explained the resampling method in detail.

Finally, for higher resolution data, we gridded the dataset to 0.05° by averaging all pixels whose center fell within each 0.05° grid cell for upscaling. For lower resolution data, we used the nearest neighbor resampling to 0.05°. In addition, MODIS data were aggregated to a monthly scale to ensure spatio-temporal consistency.

*Section 2.5: Why not utilize all available Fluxnet sites for validation instead of limiting to only Chinese sites? Would this not lead to a smaller dataset and reduce the representativeness for validating a global product?*

**REPLY:** Thanks for your comments. we have added the results of validation of GPP datasets and ensemble models using FLUXNET data in the revised version. Similarly, we extracted 0.05° MODIS land use covering the flux tower and used the site for analysis when the vegetation types of the flux tower were consistent with MODIS land use. In the end, 52 sites from FLUXNET were used. As shown in Figure R6, the validation results of the ensemble model are significantly better than those of other GPP datasets. However, underestimation is shown in the high value, which may be due to the inconsistency between the 0.05° coarse resolution and the flux tower footprint. In the revised version, we have added a description of the relevant content in the results section.

[Figure]

Figure R6. Comparison between the GPP datasets and the GPP observations from FLUXNET. a-h represents BESS, FLUXCOM, GOSIF, MODIS, NIRv, VPM, Revise-EC-LUE, ERF_GPP, respectively.

*Figures 1 and 2: It's recommended to include units for GPP, and RMSE should also specify units.*
**REPLY:** Thanks for your comments. In the revised version, we have added units.
*Figure 3: Adding seasonal variation for representative sites of different vegetation types could better highlight the model's advantages.*
**REPLY:** Thanks for your comments. In the revised version, we have added two typical sites to illustrate that the ensemble model's improvements to GPP are improvements to time series. We did not select a typical site analysis for all vegetation types because the ensemble model showed similar improvements for most sites.
As shown in Figure R7 and R8, we show the simulation results of each model at the two sites. It is obvious that $GPP_{EC}$, $GPP_{REC}$ and $GPP_{MODIS}$ on CN-Qia show obvious underestimation during the growing season. On CH_Lae, $GPP_{kNDVI}$ and $GPP_{VPM}$ are significantly overestimated. In contrast, at both sites, $GPP_{ERF}$ is more consistent with observations, meaning that the good performance of $GPP_{ERF}$ is due to the correction on the time series (although it is not well calibrated at all sites). The performance of each model is different at different sites, mainly because the process concerned by each model (environmental constraints) is different. For example, NIRv and kNDVI do not use environmental constraints in the modeling process, while other models add some constraints such as temperature. In the revised version, we have added the results of this section:
Further presentations were made at two typical sites, it was obvious that $GPP_{EC}$, $GPP_{REC}$ and $GPP_{MODIS}$ on CN-Qia showed obvious underestimation during the growing season (Figure S4). On CH_Lae, $GPP_{kNDVI}$ and $GPP_{VPM}$ were significantly overestimated (Figure S5). In contrast, at both sites, $GPP_{ERF}$ was more consistent with observations, meaning that the good performance of $GPP_{ERF}$ was due to the correction on the time series (although it was not well corrected at all sites).

[Figure]

Figure R7. Performance of each GPP model on CN-Qia.

[Figure]

Figure R8. The performance of each GPP model on CH_Lae.

*L228: Does ERF_GPP refer to the global product, while GPPERF denotes site estimation values?*

**REPLY:** Thanks for your comments. As you said, $GPP_{ERF}$ represents the site simulation and ERF_GPP represents the global GPP. In the revised version, we defined these.

*L257, NIRV should be corrected to NIRv.*

**REPLY:** Thanks for your comments. We have corrected this error in the revised version.

*In Figure S4, discrepancies with Figure 6 are noted. Is it reasonable to directly average accuracy across various sites, given differences in data quantity and the range of GPP values at different sites?*

**REPLY:** Thanks for your comments. This average is indeed not very reasonable, in the revised version, we deleted this part of the content.

*L275: What does "representative" refer to in this context?*

**REPLY:** Thanks for your comments. In the ERF model, we performed a feature importance analysis (Figure R9). From the average of 200 times, the results of the ensemble model do not depend on a single GPP simulation. Even the $GPP_{MODIS}$ with the highest relative importance does not exceed 25%, and it looks more like a weighted average of multiple GPP simulations. There is no mechanism for machine learning, so we do not know the specific reason for this result. Therefore, the term

"representative" here refers to the multiple GPP simulations, not a single GPP simulation. In the revised version, we have added a description of the relevant content in the discussion.

To further clarify the impact of explanatory variables on the ERF model, we conducted a feature importance analysis (Figure S9). From an average of 200 times, the results of the ERF model did not depend on a single GPP simulation. Even $GPP_{MODIS}$, which had the highest relative importance, was no more than 25%, so it looks more like a weighted average of multiple GPP simulations.

[Figure]

Figure R9. Average of 200 feature importance in the ERF model.

*L280-282: Some models and products already utilize dynamic temperature parameters, which the authors have not mentioned or compared.*

**REPLY:** Thanks for your comments. After searching, we found relevant study. We cite this study in the revised version and show that this refinement has the potential to improve global GPP estimates.

Previous study has shown that the estimation of GPP can be effectively improved by using dynamic temperature parameters (Chang et al., 2021).

*L283-293: Could the overestimation of low values be due to scale issues, even at the site scale, considering the used LAI is 500 m?*

**REPLY:** Thanks for your comments. The LAI of 500m is actually quite consistent with the range of the flux tower. It is possible to attribute the problem of overestimation of low values to scale problems, that is, modeling with 30m or 100m data may not have this problem. However, 30m and 100m are not in line with the observation range of the flux tower, and we believe that the modeling results under real conditions (although LAI of 500m itself is uncertain) are more reliable, that is, the high underestimation is attributed to the problem of the model structure.

*In the ERF model, is it possible to output the importance of different models during the estimation process?*

**REPLY:** Thanks for your comments. As mentioned above, the results of the ensemble model do not depend on a single GPP simulation.

*Section 4.2: Supplementing the spatial distribution of product uncertainty is recommended.*

**REPLY:** Thanks for your comments. According to your comments, we have added the spatial distribution of the uncertainty of ERF_GPP. The uncertainty of ERF_GPP mainly comes from two aspects, one is the influence of the number of GPP observations, and the other is the influence of the number of features (that is, the simulated GPP) used in the modeling process. For the first uncertainty, we randomly selected 80% of the data to build a model and simulate the multi-year average of global GPP. The process was repeated 100 times, and 100 groups of multi-year averages of ERF_GPP were obtained. Their standard deviations were considered to be the uncertainty of ERF_GPP caused by the number of GPP observations. For the second uncertainty, we choose different number of features to build models and simulate the multi-year average of global GPP. A total of 56 groups of multi-year averages of ERF_GPP are obtained. The standard deviation of different combinations is considered to be the uncertainty of ERF_GPP caused by the number of features. R10 and R11 show two types of uncertainty of ERF_GPP, similar to the spatial distribution, and ERF_GPP shows high uncertainty in the tropical regions, which has been reported in previous studies. There are very few observations of flux in these regions, both in terms of annual totals and long-term trends, and tropical regions are currently the most controversial areas in global GPP estimates. In addition, the problem of cloud pollution in remote sensing data in the tropics is well known, which further exacerbates the uncertainty in GPP estimates for the regions. In the revision, we have added a description of the relevant content and discussed it.

**2.5 Global GPP estimation based on ERF model and its uncertainty.**

Based on site-scale models, we estimated global GPP for 2001-2022 using ERF model (ERF_GPP). The uncertainty of ERF_GPP mainly comes from two aspects, one is the influence of the number of GPP observations, and the other is the influence of the number of features (that is, the simulated GPP) used in the modeling process. For the first uncertainty, we randomly selected 80% of the data to build a model and simulate the multi-year average of global GPP. The process was repeated 100 times, and 100 groups of multi-year averages of ERF_GPP were obtained. Their standard deviations were considered to be the uncertainty of ERF_GPP caused by the number of GPP observations. For the second uncertainty, we choose different number of features to build models and simulate the multi-year average of global GPP. A total of 56 groups of multi-year averages of ERF_GPP are obtained. The standard deviation of different combinations is considered to be the uncertainty of ERF_GPP caused by the number of features.

The results of the two uncertainty analyses consistently show that ERF_GPP presents a high uncertainty in the tropical region (Figure S6 and S7), and the uncertainty of ERF caused by the number of GPP observations is relatively small, the standard

deviation of 100 simulations is about 0.3 gC m$^{-2}$ d$^{-1}$ in the tropics and lower in other regions, below 0.1 gC m$^{-2}$ d$^{-1}$. In contrast, the ERF caused by the number of features is much more uncertain, especially if the number of features is small. It is worth noting that when the number of features is 5, the uncertainty is already substantially less, and the standard deviation is generally lower than 0.5 gC m$^{-2}$ d$^{-1}$.

ERF_GPP showed high uncertainty in the tropical regions, similar reports have been made in previously published GPP datasets (Badgley et al., 2019; Guo et al., 2023). There are very few observations of flux in these regions, so both in terms of annual totals and long-term trends, tropical regions are currently the most controversial areas in global GPP estimates. In addition, the problem of cloud pollution in remote sensing data in the tropics is well known (Badgley et al., 2019), which further exacerbates the uncertainty in GPP estimates for the regions.

[Figure]

Figure R10. Uncertainty of ERF_GPP caused by the number of GPP observations.

[Figure]

Figure R11. Uncertainty of ERF_GPP due to the number of features (simulated GPP).

---

## Author Response (AR2)

*Comments from the Reviewers:*

*Reviewer #1 (Formal Review for Authors):*

*In my initial assessment, I raised concerns about the evaluation as the models "saw" the full FLUXNET data. Unfortunately, this concern has not been adequately addressed, possibly due to a lack of clarity in my communication. My main concern is that the RF algorithm may have an inherent advantage over the other models insofar that the authors selected the training data randomly, i.e. most of the data of a site goes into the training data while some into the validation. In simple words, when the RF model aims to predict e.g. the November 2010 value of site 1, it might just predict the average November value of this site (temporal autocorrelation). To address this issue and increase confidence in the performance of the ensemble model, I propose the following analyses:*

**REPLY:** Thanks for your comments. We appreciate your time and effort. We are sorry that We did not fully understand your meaning in the first revision. First, the good performance of the RF model is related to the time change, which is mainly due to the seasonal change of vegetation. However, we do not consider our simulation results to be related to temporal autocorrelation. In RF models, estimates of GPP depend heavily on the characteristics of the inputs. In the initial modeling learning, GPP has established a good relationship with these features (because GPP, LAI and meteorological data have similar seasonal changes), that is, GPP=f (LAI,T,P). In the validation set, when there is a low (high) LAI input, the GPP estimate will also be low (high). That is, when we predict the value of site 1 in November 2010, the predicted value is actually determined by LAI, T, P, not just a multi-year mean. To further address your concerns, we also performed the analysis you presented.

*- How does the author's ensemble model compare to the prediction skill of a very simple RF model which only uses longitude, latitude, month, and possibly year as predictors? This comparison would clarify whether the complexity of the ensemble model significantly improves prediction skill compared to a simpler approach.*

**REPLY:** Thanks for your comments. As mentioned above, RF models that only consider longitude, latitude, year, and month are also able to achieve good simulation accuracy due to seasonal changes in vegetation (Figure R1a). In the importance analysis, we find that the importance of the month is as high as 64% (Figure R1b). However, this approach is not advisable. Because our goal in modeling at the site scale is to obtain the spatial distribution of GPP at the global or regional scale. Without the addition of remote sensing and meteorological data, the spatial distribution of GPP obtained by this simple model is completely unreliable (Figure R1c). Therefore, in the current GPP simulation, it is only meaningful to compare the simulation performance of different GPP models when remote sensing (and meteorological data) are added. In addition, we further demonstrate the simulation accuracy of this simple model at two sites (GPP$_{test}$ in Figures R2 and R3). Due to the absence of remote sensing and meteorological data, the model simulates a very small difference in the inter-annual variation and the seasonal variation (because the model

is mainly driven by months), so it is actually similar to the case where the simulated results are an average as you mentioned.

[Figure]

**Figure R1.** Simulation performance of a random forest model with only longitude, latitude, year, and month. a represents the result of the 5-fold-cross-validation, b represents the relative importance of the random forest model, and c is the multi-year mean estimated by the model for 2001-2022.

[Figure]

**Figure R2.** Performance of each GPP model on CN-Qia. The last one is a simple random forest model with only longitude, latitude, year and month.

[Figure]

**Figure R3.** The performance of each GPP model on CH_Lae. The last one is a simple random forest model with only longitude, latitude, year and month.

*- Use all the data from each site exclusively for either training (~70%) or validation (30%), instead of employing random splits. Then repeat the model creation and check model performances only on the 30% of validation subset.*

**REPLY:** Thanks for your comments. According to your suggestion, we only selected 70% of the data for training and reserve the remaining 30% for validation. This was repeated 200 times, as shown in Figure R4, and the result was very similar to the result of the 5-fold-cross-validation, with GPPERF still maintaining the highest accuracy. In the revision, we have added this validation method:

In addition, we used a second validation method where 70% of the data was selected for modeling and only the remaining 30% was validated, a process that was repeated 200 times.

Combining the results of all flux sites, GPP$_{ERF}$ explained 85.1% of the monthly GPP variations, while the seven GPP models only explained 67.7%-81.5% of the monthly GPP variations (Figure 2). Another validation method also showed similar results (Figure S3).

As for the method for evaluating the ERF model, we used the 5-fold-cross-validation, which was also used in the validation of the FLUXCOM dataset (Tramontana et al., 2016). Here we reinterpret the 5-fold-cross-validation, we divide all the data into five

pieces, select four of them (80%) for modeling, then validate the remaining one (20%), and repeat this five times to get the complete validation set. In fact, the method is similar to the one you mentioned, except that we do a loop to get the full validation result, and the above method only retains 30%. Therefore, it is inevitable that the validation results of the two methods are similar.

Tramontana G, Jung M, Schwalm C R, et al. Predicting carbon dioxide and energy fluxes across global FLUXNET sites with regression algorithms[J]. Biogeosciences, 2016, 13(14): 4291-4313.

[Figure]

**Figure R4.** Validation results for each model on 30% validation sets. The black dots represent the mean of the 200 validation results, and the upper and lower boundaries represent five times the standard deviation.

*Additionally, I appreciate the authors' inclusion of FLUXCOM for comparison, but it remains unclear which FLUXCOM product was used. Was it the RF model (which I think would be most comparable) or an ensemble of several machine learning approaches? Maybe show both? I also wonder why FLUXCOM performs so good for CHINAFLUX (Fig. R4) but much worse for FLUXNET (Fig. R5)?*

**REPLY:** Thanks for your comments. In the last revision, we used an ensemble version. in the new version, we supplemented the Random forest-based dataset, and surprisingly, in the validation, we found that the random forest-based dataset (FLUXCOM-RF) performed better than the ensemble version (FLUXCOM-ENS) (Figures R5 and R6). This may be because the ensemble version only uses simple multi-model averaging and does not get good results. Of course, even with the more accurate FLUXCOM-RF, ERF_GPP is comparable. In the revision, we compared both FLUXCOM datasets.

On the second question, you can actually see that with the exception of NIRv, the accuracy of the other products in CHINAFLUX is higher than that of FLUXNET, so the difference is obviously independent of the model structure. We think this may be because in CHINAFLUX, the input data sets (some remote sensing indicators such as LAI) of these models are strongly correlated with GPP. In contrast, in FLUXNET, the relationship between LAI and GPP is much weaker (Hu et al, 2022). Of course, this is just our guess, and it's also an interesting question that could be studied further in the future.

Hu, Z., Piao, S., Knapp, A. K., Wang, X., Peng, S., Yuan, W., ... & Yu, G. (2022). Decoupling of greenness and gross primary productivity as aridity decreases. Remote Sensing of Environment, 279, 113120.

[Figure]

**Figure R5.** Comparison between the GPP datasets and the GPP observations from ChinaFlux. a-i represents BESS, FLUXCOM-ENS, FLUXCOM-RF, GOSIF, MODIS,

[Figure]

**Figure R6.** Comparison between the GPP datasets and the GPP observations from FLUXNET. a-i represents BESS, FLUXCOM-ENS, FLUXCOM-RF, GOSIF, MODIS, NIRv, VPM, Revise-EC-LUE, ERF_GPP, respectively.

*Overall, there are still areas lacking in clarity. For example, in Table R1 "GPP number" should be revised to e.g. "number of GPP models" or "number of GPP products". Which models were added in which step? Fig. R2/R3 have no x-label.*
**REPLY:** Thanks for your comments. Sorry that there are still some errors, we have further checked the full text. Table R1 is added to further illustrate the robustness of the ERF model, which is explained in the main text.
In addition, we tested the effect of the number of GPP models on the accuracy of the ERF model. As shown in Table S8, as the number of GPP in the ERF model increased, the performance gain of the model gradually decreased.

---

## Author Response (AR3)

**Comments from the Reviewers:**

**Reviewer #1 (Formal Review for Authors):**

*I appreciate the author's additional analysis. The authors now tested a very simple RF model which only uses latitude, longitude, month and year as predictors. They found that this model has a prediction skill comparable to the ERF model for FLUXNET sites but that the predicted global maps look unrealistic. The authors are right that this simple model cannot be used to make global GPP predictions. However, this was exactly the point I was trying to make: an excellent model performance for the FLUXNET sites does not guarantee this this model skill can be transferred to the global scale. Instead, the excellent model performance at site level is inherent to way RF works, which gives it an advantage to other approaches. This does not mean your global GPP maps from the ERF model are useless but the issue has to be made very clear in the paper to avoid giving an unrealistic level of confidence in the predicted maps.*

**REPLY:** Thanks for your comments. We totally agree with you. You're thoughtful. Machine learning models have an inherent advantage, which is that even if we use variables unrelated to GPP (such as months) to estimate GPP, as long as they have the same variation characteristics, the model accuracy will be high. However, as we mentioned before, the ERF model contains remote sensing information, and some vegetation indices such as LAI have a good correlation with GPP, so the global GPP estimated by this model will not be so outrageous. In the revised version, we emphasized this point in discussions.

Due to the inherent advantages of the RF method, the accuracy of the model was comparable to that of the ERF model, even if a very simple model that used longitude, latitude, month, and year as explanatory variables (Figure S11 a). However, the global GPP estimated by this model was not reliable (Figure S11 b). This means that it is unknown whether site-scale model can be fully applied to global GPP estimates. ERF model can overcome this limitation well. On the one hand, the explanatory variables used in the model are derived from GPP simulation in which contain a lot of remote sensing information, which can ensure that the global GPP estimated by the model is reliable. On the other hand, the second validation method also further shows that the ERF model has good generalization and has greater potential than other models in estimating global GPP.

*The findings from the simple model also reinforce my second suggestion which would help to report model performance metrics that would likely be more realistic for the global maps. My suggestion was to use ALL the data from a single site exclusively either for model training or for model testing, e.g. if there are 100 sites use the data from 70 (or 80) sites for model training and test the model on the remaining 30 sites. However, the authors text reads as if they again split the data randomly. Separating the data non-randomly is important to make sure the model cannot use any data from this site to make predictions for a specific site. The prediction skill in this task would then be a better indicator of the global prediction skill of the model.*

**REPLY:** Thanks for your comments. In the previous version, we used the random split. Based on your suggestion, we conducted additional analysis. We randomly selected the data of 70% sites for model training, and validated the data of 30% sites. As shown in Figure R1, $GPP_{ERF}$ still maintains optimal model accuracy, indicating that the ERF model has greater potential for estimating global GPP than other models. In the revised version, we have added this method:

In addition, we used a second validation method in which all data from 70% of the sites were selected for modeling and only all data from the remaining 30% of the sites were validated, a process that was repeated 200 times. This validation will further illustrate the generalization of the model, i.e. its potential for estimating GPP without observations.

Combining the results of all flux sites, $GPP_{ERF}$ explained 85.1% of the monthly GPP variations, while the seven GPP estimate models only explained 67.7%-81.5% of the monthly GPP variations (Figure 2). Another validation method also showed similar results, the average $R^2$ and RMSE of 200 validation results of ERF model were 0.822 and 1.68gC m$^{-2}$ d$^{-1}$, which were obviously better than other models (Figure S3).

[Figure]

**Figure R1.** Validation results for each model on all data at 30% of the sites. The black dots represent the mean of the 200 validation results, and the upper and lower boundaries represent the standard deviation.

Finally, our paper took a lot of your time and effort, and we thank you again.

---

## Author Response (AR4)

***Comments from the Reviewers:***

***Reviewer #1 (Formal Review for Authors):***

*The authors now included a paragraph discussing the implications of the simpe lon-lat-month-year model. However, I still miss a specific statement that one has to be careful not to confuse model performance at local scale to its reliability at global scale, e.g.*
*"This illustrates that an excellent model performance for the FLUXNET sites does not neccessarily imply an equivalent prediction skill in other regions."*
*Concerning the second paragraph, I suggest*
*"...its potential for estimating GPP without local observations"*
*and also here remind the reader what this method was:*
*"Another validation method in which the validation data were not selected randomly but instead sites were entirely used for either training or validation also showed similar..."*
**REPLY:** Thank you for your careful and detailed suggestion. We have fully adopted your suggestion in the revised version. Revisions have been made in the method, results and discussions.